# ROBUST WEIGHT INITIALIZATION FOR TANH NEURAL NETWORKS WITH FIXED POINT ANALYSIS

**Hyunwoo Lee[1], Hayoung Choi[1]\*, Hyunju Kim[2]\***
[1]Kyungpook National University,  [2]Korea Institute of Energy Technology
[1]{lhw908, hayoung.choi}@knu.ac.kr, [2]hjkim@kentech.ac.kr

## ABSTRACT

As a neural network's depth increases, it can improve generalization performance. However, training deep networks is challenging due to gradient and signal propagation issues. To address these challenges, extensive theoretical research and various methods have been introduced. Despite these advances, effective weight initialization methods for tanh neural networks remain insufficiently investigated. This paper presents a novel weight initialization method for neural networks with tanh activation function. Based on an analysis of the fixed points of the function $\tanh(ax)$, the proposed method aims to determine values of $a$ that mitigate activation saturation. A series of experiments on various classification datasets and physics-informed neural networks demonstrates that the proposed method outperforms Xavier initialization methods (with or without normalization) in terms of robustness across different network sizes, data efficiency, and convergence speed. Code is available at https://github.com/1HyunwooLee/Tanh-Init.

## 1 INTRODUCTION

Deep learning has significantly advanced state-of-the-art performance across various domains (Le-Cun et al., 2015; He et al., 2016). The expressivity of neural networks increases exponentially with depth, enhancing generalization performance (Poole et al., 2016; Raghu et al., 2017). However, deeper networks often face gradient and signal propagation issues (Bengio et al., 1993). These challenges have driven the development of effective weight initialization methods designed for various activation functions. Xavier initialization (Glorot & Bengio, 2010) prevents saturation in sigmoid and tanh activations, while He initialization (He et al., 2015) stabilizes variance for ReLU networks. Especially in ReLU neural networks, several weight initialization methods have been proposed to mitigate the dying ReLU problem, which hinders signal propagation in deep networks (Lu et al., 2019; Lee et al., 2024). However, to the best of our knowledge, research on initialization methods that are robust across different sizes of tanh networks is underexplored. Tanh networks commonly employ Xavier initialization (Raissi et al., 2019; Jagtap et al., 2022; Rathore et al., 2024) and are applied in various domains, such as Physics-Informed Neural Networks (PINNs) (Raissi et al., 2019) and Recurrent Neural Networks (RNNs) (Rumelhart et al., 1986), with performance often dependent on model size and initialization randomness (Liu et al., 2022).

The main contribution of this paper is the proposal of a simple weight initialization method for FeedForward Neural Networks (FFNNs) with tanh activation function. The proposed method is data-efficient and demonstrates robustness across different network sizes. Moreover, it reduces dependency on normalization techniques such as Batch Normalization (BN) (Ioffe, 2015) and Layer Normalization (LN) (Ba, 2016). As a result, it alleviates the need for extensive hyperparameter tuning, such as selecting the number of hidden layers and units, while also eliminating the computational overhead associated with normalization. The theoretical foundation of this approach is based on the fixed point of the function $\tanh(ax)$. We evaluate the proposed method on two tasks: classification and PINNs. For classification tasks, we assess its performance across various FFNN sizes using standard benchmark datasets. The results demonstrate improved validation accuracy and lower loss compared to Xavier initialization with BN or LN. For PINNs, the method exhibits robustness across diverse network sizes and demonstrates its effectiveness in solving a wide range of PDE

---

*Corresponding authors.

problems. Notably, for both tasks, the proposed method outperforms Xavier initialization in data efficiency; that is, it achieves improved performance even with limited data. Our main contributions can be summarized as follows:

- We identify the conditions under which activation values do not vanish as the neural network depth increases, using a fixed-point analysis (Section 3.1 and 3.2).
- We propose a novel weight initialization method for tanh neural networks that is robust across different network sizes and demonstrates high data efficiency (Section 3.2 and 3.3).
- We experimentally show that the proposed method is more robust across different network sizes on image benchmarks and PINNs (Section 4).
- We experimentally show that the proposed method is more data-efficient than Xavier initialization, with or without normalization, on image benchmarks and PINNs (Section 4).

## 2   RELATED WORKS

The expressivity of neural networks grows exponentially with depth, resulting in improved generalization performance (Poole et al., 2016; Raghu et al., 2017). Proper weight initialization is crucial for effectively training deep networks (Saxe et al., 2014; Mishkin & Matas, 2016). Xavier (Glorot & Bengio, 2010) and He He et al. (2015) initialization are common initialization methods typically used with tanh and ReLU activation functions, respectively. Several initialization methods have been proposed to improve the training of deep ReLU networks (Lu et al., 2019; Bachlechner et al., 2021; Zhao et al., 2022; Lee et al., 2024). However, research on weight initialization for tanh-based neural networks remains relatively limited. Nevertheless, tanh networks have gained increasing attention in recent years, particularly in applications such as PINNs, where their performance is highly sensitive to initialization randomness.

PINNs have shown promising results in solving forward, inverse, and multiphysics problems in science and engineering. (Mao et al., 2020; Shukla et al., 2020; Lu et al., 2021; Karniadakis et al., 2021; Yin et al., 2021; Bararnia & Esmaeilpour, 2022; Cuomo et al., 2022; Hanna et al., 2022; Hosseini et al., 2023; Wu et al., 2023; Zhu et al., 2024). PINNs approximate solutions to partial differential equations (PDEs) using neural networks and are trained by minimizing a loss function, typically formulated as a sum of least-squares terms incorporating PDE residuals, boundary conditions, and initial conditions. This loss is commonly optimized using gradient-based methods such as Adam (Kingma & Ba, 2014), L-BFGS (Liu & Nocedal, 1989), or a combination of both. Universal approximation theorems (Cybenko, 1989; Hornik et al., 1989; Hornik, 1991; Park et al., 2021; Guliyev & Ismailov, 2018b; Shen et al., 2022; Guliyev & Ismailov, 2018a; Maiorov & Pinkus, 1999; Yarotsky, 2017; Gripenberg, 2003) establish the theoretical capability of neural networks to approximate analytic PDE solutions. However, PINNs still face challenges in accuracy, stability, computational complexity, and tuning appropriate hyperparameters of loss terms.

To address these challenges, various improved variants of PINNs have been proposed: (1) Self-adaptive loss-balanced PINNs (lbPINNs), which automatically adjust the hyperparameters of loss terms during training (Xiang et al., 2022), (2) Bayesian PINNs (B-PINNs), designed to handle forward and inverse nonlinear problems with noisy data (Yang et al., 2021), (3) Rectified PINNs (RPINNs), which incorporate gradient information from numerical solutions obtained via the multigrid method and are specifically designed for solving stationary PDEs (Peng et al., 2022), (4) Auxiliary PINNs (A-PINNs), developed to effectively handle integro-differential equations (Yuan et al., 2022), (5) Conservative PINNs (cPINNs) and Extended PINNs (XPINNs), which employ domain decomposition techniques (Jagtap et al., 2020; Jagtap & Karniadakis, 2020), (6) Parallel PINNs, designed to reduce the computational cost of cPINNs and XPINNs (Shukla et al., 2021), and (7) Gradient-enhanced PINNs (gPINNs), which incorporate the gradient of the PDE loss term with respect to network inputs (Yu et al., 2022).

While these advancements address various challenges in PINNs, activation functions and their initialization strategies remain crucial for achieving optimal performance. The tanh activation function has been shown to perform well in PINNs (Raissi et al., 2019), with detailed experimental results provided in Appendix C.2. Xavier initialization is widely employed as the standard approach for tanh networks in existing studies (Jin et al., 2021; Son et al., 2023; Yao et al., 2023; Gnanasambandam et al., 2023; Song et al., 2024). However, our experimental results indicate that employing

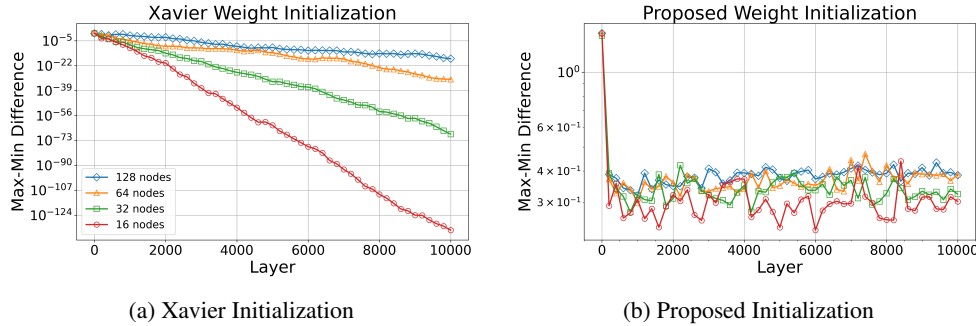

(a) Xavier Initialization     (b) Proposed Initialization

Figure 1: The difference between the maximum and minimum activation values at each layer when propagating $3,000$ input samples through a $10,000$-layer tanh FFNN, using Xavier initialization (**left**) and the proposed initialization (**right**). Experiments were conducted on distinct networks with $10,000$ hidden layers, each having the same number of nodes: $16, 32, 64,$ or $128$.

Xavier initialization leads to decreased model performance as network size increases. Moreover, performance gains from Batch Normalization or Layer Normalization remain limited, and Xavier initialization exhibits sensitivity to the amount of training data, particularly in smaller datasets. Although a recent study has proposed an initialization method for PINNs, it depends on transfer learning (Tarbiyati & Nemati Saray, 2023). To address these limitations, we propose a weight initialization method that is robust across different network sizes, achieves high data efficiency, and reduces reliance on both transfer learning and normalization techniques.

## 3 PROPOSED WEIGHT INITIALIZATION METHOD

In this section, we discuss the proposed weight initialization method. Section 3.1 introduces the theoretical motivation behind the method. Section 3.2 presents how to derive the initial weight matrix that satisfies the conditions outlined in Section 3.1. Finally, in Section 3.3, we suggest the optimal hyperparameter $\sigma_z$ in the proposed method.

### 3.1 THEORETICAL MOTIVATION

Experimental results in Figure 1 indicate that when Xavier initialization is employed in FFNNs with tanh activation, the activation values tend to vanish toward zero in deeper layers. This vanishing of activation values can hinder the training process due to a discrepancy between the activation values and the desired output. However, it is not straightforward to theoretically establish the conditions for preventing this phenomenon. In this section, we present a theoretical analysis based on a fixed point of $\tanh(ax)$ to mitigate this issue. Before giving the theoretical foundations, consider the basic results for a hyperbolic tangent activation function. Recall that if $f : \mathbb{R} \to \mathbb{R}$ is a function, then an element $x^* \in \mathbb{R}$ is called a *fixed point* of $f$ if $f(x^*) = x^*$.

**Lemma 1.** *For a fixed $a > 0$, define the function $\phi_a : \mathbb{R} \to \mathbb{R}$ given as*

$$\phi_a(x) := \tanh(ax).$$

*Then, there exists a fixed point $x^*$ of $\phi_a$. Furthermore,*

(1) *if $0 < a \leq 1$, then $\phi_a$ has a unique fixed point $x^* = 0$.*

(2) *if $a > 1$, then $\phi_a$ has three distinct fixed points: $x^* = -\xi_a, 0, \xi_a$ such that $\xi_a > 0$.*

*Proof.* The proof is detailed in Appendix A.1. $\square$

Note that $\tanh(x) < x$ for all $x > 0$. However, according to Lemma 1, for $a > 1$ the behavior of $\tanh(ax)$ changes. If $x > \xi_a$ (resp. $x < \xi_a$), then $\tanh(ax) < x$ (resp. $\tanh(ax) > x$). At $x = \xi_a$, the equality $\tanh(ax) = x$ holds. The following lemma addresses the convergence properties of iteratively applying $\tanh(ax)$ for any $x > 0$.

**Lemma 2.** *For a given initial value $x_0 > 0$, define*

$$x_{n+1} = \phi_a(x_n), \quad n = 0, 1, 2, \ldots.$$

*Then $\{x_n\}_{n=1}^{\infty}$ converges regardless of the positive initial value $x_0 > 0$. Moreover,*

    (1) *if $0 < a \leq 1$, then $x_n \to 0$ as $n \to \infty$.*

    (2) *if $a > 1$, then $x_n \to \xi_a$ as $n \to \infty$.*

*Proof.* The proof is detailed in Appendix A.2.     □

Note that for $a > 1$ and $x_0 > \xi_a > 0$, the sequence $\{x_n\}$ satisfies $\tanh(ax_n) > \tanh(ax_{n+1}) > \xi_a$ for all $n \in \mathbb{N}$. Similarly, when $0 < x_0 < \xi_a$, the sequence satisfies $\tanh(ax_n) < \tanh(ax_{n+1}) < \xi_a$ for all $n \in \mathbb{N}$. Given $a > 1$ and $x_0 < 0$, the sequence converges to $-\xi_a$ as $n \to \infty$ due to the odd symmetry of $\tanh(ax)$. According to Lemma 2, when $a > 1$, the sequence $\{x_n\}$ converges to $\xi_a$ or $-\xi_a$, respectively, as $n \to \infty$, for an arbitrary initial value $x_0 > 0$ or $x_0 < 0$.

Note that the parameter $a$ in Lemma 2 does not change across all iterations. Propositions 3 and Corollary 4, consider the case where the value of $a$ varies for each iteration.

**Proposition 3.** *Let $\{a_n\}_{n=1}^{\infty}$ be a positive real sequence, i.e., $a_n > 0$ for all $n \in \mathbb{N}$, such that only finitely many elements are greater than 1. Suppose that $\{\Phi_m\}_{m=1}^{\infty}$ is a sequence of functions defined as for each $m \in \mathbb{N}$*

$$\Phi_m = \phi_{a_m} \circ \phi_{a_{m-1}} \circ \cdots \circ \phi_{a_1}.$$

*Then for any $x \in \mathbb{R}$*

$$\lim_{m \to \infty} \Phi_m(x) = 0.$$

*Proof.* The proof is detailed in Appendix A.3.     □

**Corollary 4.** *Let $\epsilon > 0$ be given. Suppose that $\{a_n\}_{n=1}^{\infty}$ be a positive real sequence such that only finitely many elements are lower than $1 + \epsilon$. Then for any $x \in \mathbb{R} \setminus \{0\}$*

$$\lim_{m \to \infty} |\Phi_m(x)| \geq \xi_{1+\epsilon}.$$

*Proof.* The proof is detailed in Appendix A.4.     □

Based on Proposition 3 and Corollary 4, if there exists a sufficiently large $N \in \mathbb{N}$ such that $a_n < 1$ (resp. $a_n > 1 + \epsilon$) for all $n \geq N$, then for any $x_0 \in \mathbb{R} \setminus \{0\}$, $\Phi_m(x_0) \to 0$ (resp. $|\Phi_m(x_0)| \geq \xi_{1+\epsilon}$) as $m \to \infty$. This result implies that if the sequence $\{\Phi_m\}_{m=1}^{M}$ is finite, $a_n$ for $N \leq n \leq M$, where $N$ is an arbitrarily chosen index close to $M$, significantly influence the values of $\Phi_M(x_0)$.

## 3.2 The derivation of the proposed weight initialization method

Based on the theoretical motivations discussed in the previous section, we propose a weight initialization method that satisfies the following conditions in the initial forward pass:

    (i) It prevents activation values from vanishing towards zero in deep layers.

    (ii) It ensures that the distribution of activation values in deep layers is approximately normal.

**Notations.** Consider a feedforward neural network with $L$ layers. The network processes $K$ training samples, denoted as pairs $\{(\boldsymbol{x}_i, \boldsymbol{y}_i)\}_{i=1}^{K}$, where $\boldsymbol{x}_i \in \mathbb{R}^{N_x}$ is training input and $\boldsymbol{y}_i \in \mathbb{R}^{N_y}$ is its corresponding output. The iterative computation at each layer $\ell$ is defined as follows:

$$\boldsymbol{x}^{\ell} = \tanh(\mathbf{W}^{\ell}\boldsymbol{x}^{\ell-1} + \mathbf{b}^{\ell}) \in \mathbb{R}^{N_\ell} \quad \text{for all } \ell = 1, \ldots, L,$$

where $\mathbf{W}^{\ell} \in \mathbb{R}^{N_\ell \times N_{\ell-1}}$ is the weight matrix, $\mathbf{b}^{\ell} \in \mathbb{R}^{N_\ell}$ is the bias, and $\tanh(\cdot)$ is an element-wise activation hyperbolic tangent function. Denote $\mathbf{W}^{\ell} = [w_{ij}^{\ell}]$.

**Signal Propagation Analysis.** We present a simplified analysis of signal propagation in FFNNs with the tanh activation function. For notational convenience, it is assumed that all hidden layers, as well

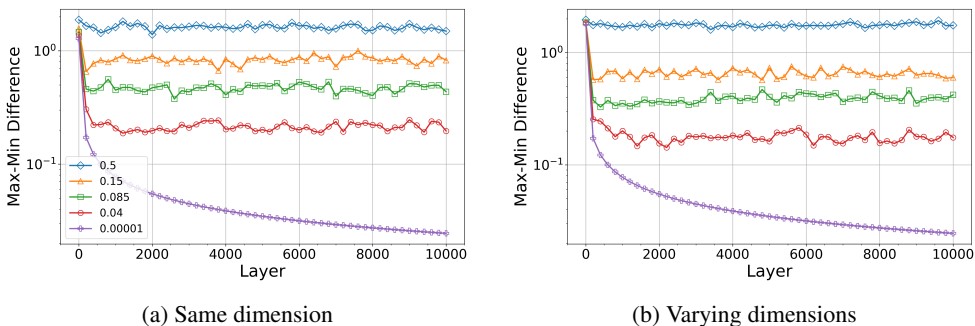

Figure 2: The difference between the maximum and minimum activation values at each layer when propagating $3,000$ input samples through a $10,000$-layer tanh FFNN, using the proposed initialization with $\alpha$ set to $0.04, 0.085, 0.15$, and $0.5$. Network with $10,000$ hidden layers, each with 32 nodes **(left)**, and a network with alternating hidden layers of 64 and 32 nodes **(right)**.

as the input and output layers, have a dimension of $n$, i.e., $N_\ell = n$ for all $\ell$. Given an arbitrary input vector $\boldsymbol{x} = (x_1, \ldots, x_n)$, the first layer activation $\boldsymbol{x}^1 = \tanh(\mathbf{W}^1 \boldsymbol{x})$ can be expressed component-wise as

$$x_i^1 = \tanh\left(w_{i1}^1 x_1 + \cdots + w_{in}^1 x_n\right) = \tanh\left(\left(w_{ii}^1 + \sum_{\substack{j=1 \\ j \neq i}}^{n} \frac{w_{ij}^1 x_j}{x_i}\right) x_i\right) \text{ for } i = 1, \ldots, n.$$

For the $(k+1)$-th layer this expression can be generalized as

$$x_i^{k+1} = \tanh\left(a_i^{k+1} x_i^k\right), \text{ where } \quad a_i^{k+1} = w_{ii}^{k+1} + \sum_{\substack{j=1 \\ j \neq i}}^{n} \frac{w_{ij}^{k+1} x_j^k}{x_i^k} \quad \text{ for } i = 1, \ldots, n. \tag{1}$$

Equation (1) follows the form of $\tanh(ax)$, as discussed in Section 3.2. According to Lemma 2, when $a > 1$, for an arbitrary initial value $x_0 > 0$ or $x_0 < 0$, the sequence $\{x_k\}$ defined by $x_{k+1} = \tanh(ax_k)$ converges to $\xi_a$ or $-\xi_a$, respectively, as $k \to \infty$. This result indicates that the sequence converges to the fixed point $\xi_a$ regardless of the initial value $x_0$. From the perspective of signal propagation in tanh-based FFNNs, this ensures that the activation values do not vanish as the network depth increases. Furthermore, by Proposition 3, if $a_i^k \leq 1$ for all $N \leq k \leq L$, where $N$ is an arbitrarily chosen index sufficiently close to $L$, the value of $x_i^L$ approaches zero. Therefore, to satisfy condition (i), $a_i^k$ remains close to 1, and the inequality $a_i^k \leq 1$ does not hold for all $N \leq k \leq L$.

**Proposed Weight Initialization.** The proposed initial weight matrix is defined as $\mathbf{W}^\ell = \mathbf{D}^\ell + \mathbf{Z}^\ell \in \mathbb{R}^{N_\ell \times N_{\ell-1}}$, where $\mathbf{D}_{i,j}^\ell = 1$ if $i \equiv j \pmod{N_{\ell-1}}$, and 0 otherwise (Examples of $\mathbf{D}^\ell$ are provided in Appendix D). The noise matrix $\mathbf{Z}^\ell$ is drawn from $\mathcal{N}(0, \sigma_z^2)$, where $\sigma_z$ is set to $\alpha/\sqrt{N^{\ell-1}}$ with $\alpha = 0.085$. Then $a_i^{k+1}$ follows the distribution:

$$a_i^{k+1} \sim \mathcal{N}\left(1, \sigma_z^2 + \sigma_z^2 \sum_{\substack{j=1 \\ j \neq i}}^{n} \left(\frac{x_j^k}{x_i^k}\right)^2\right). \tag{2}$$

According to Equation (2), $a_i^{k+1}$ follows a Gaussian distribution with a mean of 1. Additionally, if $x_i^k$ becomes small relative to other elements in $\boldsymbol{x}^k$, the variance of the distribution in (2) increases. Consequently, the probability that the absolute value of $x_i^{k+1}$ exceeds that of $x_i^k$ becomes higher. Figure 1 (b) shows that activation values maintain consistent scales in deeper layers.

### 3.3 PREVENTING ACTIVATION SATURATION VIA APPROPRIATE $\sigma_z$ TUNING

In this section, we determine the appropriate value of $\alpha$ in $\sigma_z = \alpha/\sqrt{N_{\ell-1}}$ that satisfies condition (ii). Condition (ii) is motivated by normalization methods (Ioffe, 2015; Ba, 2016). Firstly,

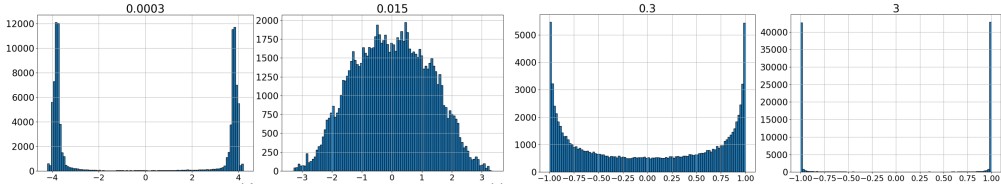

Figure 3: The activation values in the 1000$^{\text{th}}$ layer, with 32 nodes per hidden layer, were analyzed using the proposed weight initialization method with $\sigma_z$ values of 0.0003, 0.015, 0.3, and 3. The analysis was conducted on 3, 000 input samples uniformly distributed within the range $[-1, 1]$.

we experimentally investigated the impact of $\sigma_z$ on the scale of the activation values. As shown in Figure 2, increasing $\sigma_z = \alpha/\sqrt{N_{\ell-1}}$ broadens the activation range in each layer, while decreasing $\sigma_z$ narrows it.

**When $\sigma_z$ is Large.** Setting $\sigma_z$ to a large value can lead to saturation. If $\sigma_z$ is too large, Equation (2) implies that the likelihood of $a_i^k$ deviating significantly from 1 increases. This increases the likelihood of activation values being bounded by $\xi_{1+\epsilon}$ in sufficiently deep layers, as stated in Corollary 4. Consequently, in deeper layers, activation values are less likely to approach zero and tend to saturate toward specific values. Please refer to the Figure 3 for the cases where $\sigma_z = 0.3$ and 3.

**When $\sigma_z$ is Small.** If $\sigma_z$ is too small, Equation (2) implies that the distribution of $a_i^k$ has a standard deviation close to zero. Consequently, $x_i^{k+1}$ can be approximated as the result of applying $\tanh(x)$ to $x_i > 0$ repeatedly for a finite number of iterations, $k$. Since $\tanh'(x)$ decreases for $x \geq 0$, the values resulting from finite iterations eventually saturate. Plese refer to the Figure 3 when $\sigma_z = 0.0003$.

For these reasons, we experimentally determined an optimal $\sigma_z$ that avoids being excessively large or small. As shown in Figure 3, $\sigma_z = 0.015$ maintains an approximately normal activation distribution without collapse. Additional experimental results are provided in Appendix A.5. Considering the number of hidden layer nodes, we set $\sigma_z = \alpha/\sqrt{N^{\ell-1}}$ with $\alpha = 0.085$. Experimental results for solving the Burgers' equation using PINNs with varying $\sigma_z$ are provided in Appendix C.3.

## 4 EXPERIMENTS

In this section, we present a series of experiments to evaluate the proposed weight initialization method. In Section 4.1, we evaluate the performance of tanh FFNNs on benchmark classification datasets. In Section 4.2, we apply Physics-Informed Neural Networks to solve PDEs. Both experiments assess the proposed method's robustness to network size and data efficiency.

Table 1: Validation accuracy and loss are presented for FFNNs with varying numbers of nodes $(2, 8, 32, 128, 512)$, each with 20 hidden layers using tanh activation function. All models were trained for 20 epochs, and the highest average accuracy and lowest average loss, computed over 10 runs, are presented. The better-performing method is highlighted in bold when comparing different initialization methods under the same experimental settings.

| Dataset | Method | 2 Nodes | | 8 Nodes | | 32 Nodes | | 128 Nodes | | 512 Nodes | |
|---|---|---|---|---|---|---|---|---|---|---|---|
| | | Accuracy | Loss | Accuracy | Loss | Accuracy | Loss | Accuracy | Loss | Accuracy | Loss |
| MNIST | Xavier | 49.78 | 1.632 | 68 | 0.958 | 91.67 | 0.277 | 95.45 | 0.154 | 97.35 | 0.087 |
| | Proposed | **62.82** | **1.185** | **77.95** | **0.706** | **92.51** | **0.255** | **96.12** | **0.134** | **97.96** | **0.067** |
| FMNIST | Xavier | 42.89 | 1.559 | 68.55 | 0.890 | 81.03 | 0.533 | 86.20 | 0.389 | 88.28 | 0.331 |
| | Proposed | **51.65** | **1.324** | **71.31** | **0.777** | **83.06** | **0.475** | **87.12** | **0.359** | **88.59** | **0.323** |
| CIFAR-10 | Xavier | 32.82 | 1.921 | 43.51 | 1.608 | 48.62 | 1.473 | 47.58 | 1.510 | 51.71 | 1.369 |
| | Proposed | **38.16** | **1.780** | **47.04** | **1.505** | **48.80** | **1.463** | **48.51** | **1.471** | **52.21** | **1.359** |
| CIFAR-100 | Xavier | 10.87 | 4.065 | 18.53 | 3.619 | 23.71 | 3.301 | 23.83 | 3.324 | 17.72 | 3.672 |
| | Proposed | **15.22** | **3.818** | **23.07** | **3.350** | **24.93** | **3.237** | 24.91 | **3.240** | 22.80 | **3.435** |

Table 2: Validation accuracy and loss are presented for FFNNs with varying numbers of layers $(3, 10, 50, 100)$, each with $64$ number of nodes using the tanh activation function. All models were trained for $40$ epochs, and the highest average accuracy and lowest average loss, computed over $10$ runs, are presented.

| Dataset | Method | 3 Layers | | 10 Layers | | 50 Layers | | 100 Layers | |
|---|---|---|---|---|---|---|---|---|---|
| | | Accuracy | Loss | Accuracy | Loss | Accuracy | Loss | Accuracy | Loss |
| MNIST | Xavier | 95.98 | 0.130 | 96.55 | 0.112 | 96.57 | 0.123 | 94.08 | 0.194 |
| | Proposed | **96.32** | **0.123** | **97.04** | **0.102** | **96.72** | **0.109** | **96.06** | **0.132** |
| FMNIST | Xavier | 85.91 | 0.401 | 88.73 | 0.319 | 87.72 | 0.344 | 83.41 | 0.463 |
| | Proposed | **86.51** | **0.379** | **89.42** | **0.305** | **88.51** | **0.324** | **86.01** | **0.382** |
| CIFAR-10 | Xavier | 42.91 | 1.643 | 48.39 | 1.468 | 47.87 | 1.474 | 46.71 | 1.503 |
| | Proposed | **45.05** | **1.588** | **48.41** | **1.458** | **48.71** | **1.461** | **48.96** | **1.437** |
| CIFAR-100 | Xavier | 19.10 | 3.628 | 22.73 | 3.400 | 24.27 | 3.283 | 20.32 | 3.515 |
| | Proposed | **19.30** | **3.609** | **23.83** | **3.309** | **25.07** | **3.190** | **24.41** | **3.234** |

## 4.1 CLASSIFICATION TASK

**Experimental Setting.** To evaluate the effectiveness of the proposed weight initialization method, we conduct experiments on the MNIST, Fashion MNIST (FMNIST), CIFAR-10, and CIFAR-100 (Krizhevsky & Hinton, 2009) datasets with the Adam optimizer. All experiments are performed with a batch size of $64$ and a learning rate of $0.0001$. Fifteen percent of the dataset is allocated for validation. The experiments are implemented in TensorFlow without skip connections and learning rate decay in any of the experiments.

**Width Independence in Classification Task.** We evaluate the proposed weight initialization method in training tanh FFNNs, focusing on its robustness across different network widths. Five tanh FFNNs are designed, each with 20 hidden layers, and with $2, 8, 32, 128,$ and $512$ nodes per hidden layer, respectively. In Table 1, for the MNIST, Fashion MNIST and CIFAR-10 datasets, the network with 512 nodes achieves the highest accuracy and lowest loss when the proposed method is employed. However, for the CIFAR-100 dataset, the network with 32 nodes yields the highest accuracy and lowest loss when employing the proposed method. In summary, the proposed method demonstrates robustness across different network widths in tanh FFNNs. Detailed experimental results are provided in Appendix B.1.

**Depth Independence in Classification Task.** The expressivity of neural networks is known to increase exponentially with depth, leading to improved generalization performance (Poole et al., 2016; Raghu et al., 2017). To evaluate the robustness of the proposed weight initialization method across different network depths, we conduct experiments on deep FFNNs with tanh activation functions. Specifically, we construct four tanh FFNNs, each with 64 nodes per hidden layer and 3, 10, 50, and 100 hidden layers. respectively. In Table 2, for both the MNIST and Fashion MNIST datasets, the network with 10 hidden layers achieves the highest accuracy and lowest loss when our proposed method is employed. Both initialization methods exhibit lower performance in networks with 3 layers compared to those with more layers. Moreover, for more complex datasets such as CIFAR-10 and CIFAR-100, the proposed method demonstrated improved performance when training deeper networks.

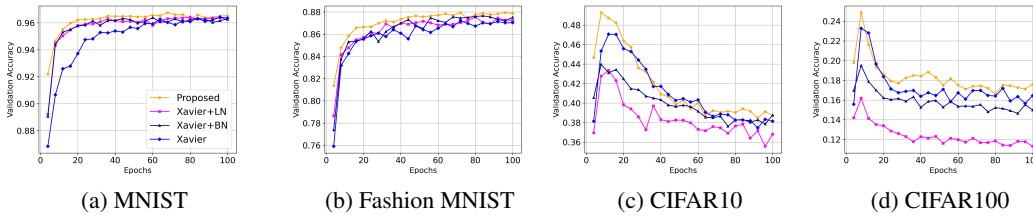

|  (a) MNIST | (b) Fashion MNIST | (c) CIFAR10 | (d) CIFAR100 |

Figure 4: Validation accuracy for a tanh FFNN with 50 hidden layers (32 nodes each). Xavier + BN and Xavier + LN represent Xavier initialization with Batch Normalization or Layer Normalization applied every 5 layers, respectively.

Table 3: Validation accuracy and loss for a 10-layer FFNN (64 nodes per layer) trained on datasets containing 10, 20, 30, 50, and 100 samples. Results show the highest average accuracy and lowest average loss over 5 runs after 100 epochs.

| Dataset | Method | 10 | | 20 | | 30 | | 50 | | 100 | |
|---------|--------|----------|-------|----------|-------|----------|-------|----------|-------|----------|-------|
| | | Accuracy | Loss | Accuracy | Loss | Accuracy | Loss | Accuracy | Loss | Accuracy | Loss |
| MNIST | Xavier | 31.13 | 2.281 | 35.03 | 2.078 | 45.05 | 1.771 | 58.45 | 1.227 | 64.02 | 1.139 |
| | Xavier + BN | 22.46 | 2.267 | 33.73 | 2.053 | 37.13 | 2.042 | 39.78 | 1.944 | 57.51 | 1.464 |
| | Xavier + LN | 28.52 | 2.411 | 41.54 | 1.796 | 41.94 | 1.886 | 54.97 | 1.362 | 65.11 | 1.093 |
| | Proposed | **37.32** | **2.204** | **46.79** | **1.656** | **48.60** | **1.645** | **61.54** | **1.131** | **68.44** | **1.043** |
| FMNIST | Xavier | 36.16 | 2.320 | 41.69 | 1.814 | 53.86 | 1.459 | 64.53 | 1.140 | 63.58 | 1.048 |
| | Xavier + BN | 35.44 | **2.136** | 38.58 | 1.925 | 40.16 | 1.819 | 53.93 | 1.728 | 59.78 | 1.237 |
| | Xavier + LN | 34.94 | 2.362 | 37.90 | 1.793 | 53.27 | 1.470 | 59.50 | 1.198 | 62.01 | 1.073 |
| | Proposed | **37.31** | 2.217 | **49.25** | **1.651** | **55.19** | **1.372** | **66.14** | **1.057** | **67.58** | **0.914** |

**Normalization Methods.** Xavier initialization is known to cause vanishing gradients and activation problems in deeper networks. These issues can be mitigated by applying Batch Normalization (BN) or Layer Normalization (LN) in the network. Therefore, we compare the proposed method with Xavier, Xavier with BN, and Xavier with LN. To evaluate the effectiveness of normalization, we conducted experiments using a sufficiently deep neural network with 50 hidden layers. As shown in Figure 4, for datasets with relatively fewer features, such as MNIST and FMNIST, Xavier with normalization converges faster than Xavier. However, for feature-rich datasets such as CIFAR-10 and CIFAR-100, the accuracy of Xavier with normalization is lower than that of Xavier. Normalization typically incurs a 30% computational overhead, and additional hyperparameter tuning is required to determine which layers should apply normalization. In contrast, the proposed method achieves the best performance across all datasets without requiring normalization.

**Data Efficiency in Classification Task.** Based on the results in Table 2, we evaluated data efficiency on a network with 50 hidden layers, each containing 64 nodes, where Xavier exhibited strong performance. As shown in Table 3, the highest average accuracy and lowest average loss over 5 runs after 100 epochs are presented for datasets containing 10, 20, 30, 50, and 100 samples. The proposed method achieved the best performance across all sample sizes.

**Non-uniform Hidden Layer Dimensions.** We evaluate the performance of the proposed initialization in networks where hidden layer dimensions are not uniform. As shown in Figure 5, the network consists of 60 hidden layers, where the number of nodes alternates between 32 and 16 in each layer. The proposed method demonstrates improved performance in terms of both loss and accuracy across all epochs on the MNIST and CIFAR-10 datasets. Additionally, Appendix B.2 presents experiments on networks with larger variations in the number of nodes. Motivated by these results, Appendix B.2 further explores autoencoders with significant differences in hidden layer dimensions.

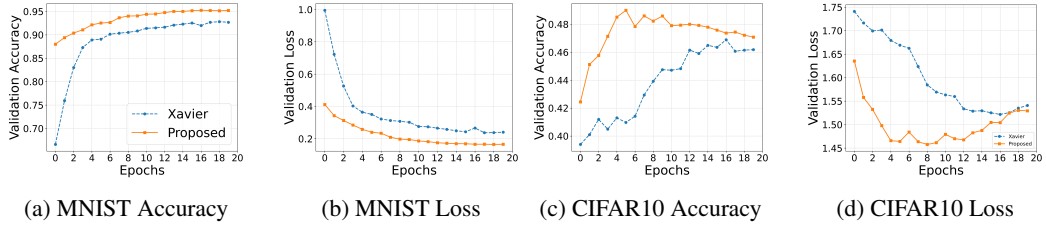

| (a) MNIST Accuracy | (b) MNIST Loss | (c) CIFAR10 Accuracy | (d) CIFAR10 Loss |
|---|---|---|---|

Figure 5: Validation accuracy and loss for a tanh FFNN with 60 hidden layers, where the number of nodes alternates between 32 and 16 across layers, repeated 30 times. The model was trained for 20 epochs on the MNIST and CIFAR-10 datasets.

## 4.2 Physics-Informed Neural Networks

Xavier initialization is the most commonly employed method for training PINNs (Jin et al., 2021; Son et al., 2023; Yao et al., 2023; Gnanasambandam et al., 2023). In this section, we experimentally demonstrate that the proposed method is more robust across different network sizes and achieves higher data efficiency compared to Xavier initialization with or without normalization methods.

**Experimental Setting.** All experiments on Physics-Informed Neural Networks (PINNs) use full-batch training with a learning rate of 0.001. In this section, we solve the Allen-Cahn, Burgers, Diffusion, and Poisson equations using a tanh FFNN-based PINN with 20,000 collocation points. Details on the PDEs are provided in Appendix C.1.

**Network Size Independence in PINNs.** We construct eight tanh FFNNs, each with 16 nodes per hidden layer and $5, 10, 20, 30, 40, 50, 60,$ or $80$ hidden layers. As shown in Table 4, for the Allen-Cahn and Burgers' equations, Xavier+BN and Xavier+LN achieve the lowest loss at a network depth of 30. However, their loss gradually increases as depth grows. In contrast, the proposed method achieves the lowest loss at depths of 50 and 60, respectively, maintaining strong learning performance even in deeper networks. For the Diffusion and Poisson equations, Xavier+LN achieves the lowest loss at depths of 5 and 10, respectively. While all methods exhibit increasing loss as network depth increases, the proposed method consistently maintains lower loss in deeper networks. Similar trends are observed with 32 nodes. Across all tested network sizes and PDEs, the proposed method consistently achieves the lowest loss. The proposed method eliminates the computational overhead and hyperparameter tuning required for normalization methods.

Table 4: A PINN loss is presented for FFNNs with varying numbers of layers $(5, 10, 20, 30, 40, 50, 60, 80)$ using the tanh activation function. The top table shows results with 16 nodes per layer, and the bottom table shows results with 32 nodes per layer. All models were trained for 300 iterations using Adam and 300 iterations using L-BFGS. The median PINN loss at the final iteration for the Burgers, Allen–Cahn, Diffusion, and Poisson equations, computed over 5 runs, is presented.

| Allen-Cahn (16 Nodes) | 5 | 10 | 20 | 30 | 40 | 50 | 60 | 80 |
|---|---|---|---|---|---|---|---|---|
| Xavier | 9.58e-04 | 8.16e-04 | 7.61e-04 | 1.06e-03 | 1.1e-03 | 1.24e-03 | 3.55e-03 | 1.81e-03 |
| Xavier + BN | 1.42e-03 | 8.17e-04 | 8.56e-04 | 7.07e-04 | 7.77e-04 | 8.87e-04 | 9.11e-04 | 2.15e-03 |
| Xavier + LN | 6.29e-01 | 1.77e-03 | 6.98e-04 | 1.27e-03 | 1.82e-03 | 6.65e-01 | 3.29e-01 | 5.86e-01 |
| Proposed | **9.21e-04** | **7.29e-04** | **5.76e-04** | **5.29e-04** | **5.37e-04** | **4.03e-04** | **4.73e-04** | **5.77e-04** |

| Burgers (16 Nodes) | 5 | 10 | 20 | 30 | 40 | 50 | 60 | 80 |
|---|---|---|---|---|---|---|---|---|
| Xavier | 6.97e-03 | 1.11e-02 | 7.9e-03 | 9.71e-03 | 2.45e-02 | 2.65e-02 | 6.5e-02 | 5.71e-02 |
| Xavier + BN | 8.07e-03 | 7.72e-03 | 6.24e-03 | 1.70e-02 | 1.50e-02 | 1.85e-02 | 2.91e-02 | 6.84e-02 |
| Xavier + LN | 3.89e-02 | 1.88e-02 | 9.48e-03 | 9.28e-03 | 2.46e-02 | 3.30e-02 | 6.91e-02 | 4.42e-02 |
| Proposed | **6.19e-03** | **5.08e-03** | **5.28e-03** | **9.31e-04** | **3.56e-03** | **8.27e-04** | **3.43e-04** | **2.05e-03** |

| Diffusion (16 Nodes) | 5 | 10 | 20 | 30 | 40 | 50 | 60 | 80 |
|---|---|---|---|---|---|---|---|---|
| Xavier | 2.52e-03 | 4.82e-03 | 9.69e-03 | 1.33e-02 | 2.08e-02 | 1.50e-02 | 2.92e-02 | 7.24e-02 |
| Xavier + BN | 2.89e-03 | 5.77e-03 | 1.05e-02 | 9.65e-03 | 2.76e-02 | 1.07e-02 | 9.07e-03 | 1.43e-02 |
| Xavier + LN | 1.72e-03 | 6.10e-03 | 8.04e-03 | 9.48e-03 | 2.14e-02 | 7.59e-03 | 2.05e-02 | 2.21e-02 |
| Proposed | **9.14e-04** | **2.59e-03** | **2.40e-03** | **1.01e-03** | **1.97e-03** | **1.21e-03** | **1.12e-03** | **1.91e-03** |

| Poisson (16 Nodes) | 5 | 10 | 20 | 30 | 40 | 50 | 60 | 80 |
|---|---|---|---|---|---|---|---|---|
| Xavier | 1.52e-02 | 2.87e-02 | 1.28e-01 | 9.82e-02 | 1.15e-01 | 1.37e-01 | 1.82e-01 | 2.55e-01 |
| Xavier + BN | 1.62e-02 | 2.02e-02 | 8.72e-02 | 1.12e-01 | 2.45e-01 | 9.85e-02 | 1.00e-01 | 1.34e-01 |
| Xavier + LN | 5.39e-01 | 4.40e-02 | 1.34e-01 | 3.91 | 2.52e+02 | 2.58 | 9.79e+02 | N/A |
| Proposed | **1.37e-02** | **1.70e-02** | **4.62e-02** | **2.43e-02** | **3.75e-02** | **4.03e-02** | **6.07e-02** | **6.01e-02** |

| Allen-Cahn (32 Nodes) | 5 | 10 | 20 | 30 | 40 | 50 | 60 | 80 |
|---|---|---|---|---|---|---|---|---|
| Xavier | 3.13e-01 | 5.03e-02 | 3.64e-03 | 2.37e-03 | 4.03e-03 | 5.27e-03 | 1.73e-02 | 6.94e-01 |
| Xavier + BN | 4.05e-01 | 8.85e-04 | 8.41e-04 | 7.82e-04 | 9.97e-04 | 6.80e-04 | 9.34e-04 | 6.94e-01 |
| Xavier + LN | 3.31e-01 | 2.10e-03 | 5.99e-04 | 6.71e-04 | 1.49e-03 | 1.29e-03 | 3.31e-02 | 6.93e-01 |
| Proposed | **1.04e-03** | **6.92e-04** | **5.34e-04** | **4.26e-04** | **3.31e-04** | **3.52e-04** | **3.85e-04** | **5.96e-04** |

| Burgers (32 Nodes) | 5 | 10 | 20 | 30 | 40 | 50 | 60 | 80 |
|---|---|---|---|---|---|---|---|---|
| Xavier | 1.12e-02 | 3.53e-03 | 2.72e-03 | 1.81e-03 | 7.60e-03 | 8.56e-03 | 9.86e-03 | 1.66e-01 |
| Xavier + BN | 5.88e-03 | **1.04e-03** | 1.79e-03 | 2.80e-03 | 5.95e-03 | 3.66e-02 | 6.60e-02 | 1.66e-01 |
| Xavier + LN | 4.31e-02 | 1.21e-02 | 1.88e-03 | 7.22e-03 | 5.54e-03 | 8.46e-03 | 9.04e-03 | 4.86e-02 |
| Proposed | **4.14e-03** | 4.11e-03 | **1.58e-03** | **1.29e-03** | **7.96e-04** | **5.85e-04** | **9.80e-04** | **1.47e-03** |

| Diffusion (32 Nodes) | 5 | 10 | 20 | 30 | 40 | 50 | 60 | 80 |
|---|---|---|---|---|---|---|---|---|
| Xavier | 1.69e-03 | 6.85e-03 | 7.63e-03 | 4.50e-03 | 8.98e-03 | 5.67e-03 | 6.33e-01 | 1.59 |
| Xavier + BN | 1.68e-03 | 2.66e-03 | 1.08e-02 | 6.00e-03 | 8.58e-03 | 6.60e-03 | 5.66e-02 | 1.69e+02 |
| Xavier + LN | 8.16e-04 | 2.85e-03 | 8.46e-03 | 4.57e-03 | 9.40e-03 | 1.04e-02 | 2.42e-01 | 1.67e+02 |
| Proposed | **2.89e-04** | **8.03e-04** | **5.25e-04** | **5.07e-04** | **5.33e-04** | **6.17e-04** | **9.80e-04** | **1.53e-03** |

| Poisson (32 Nodes) | 5 | 10 | 20 | 30 | 40 | 50 | 60 | 80 |
|---|---|---|---|---|---|---|---|---|
| Xavier | 1.09e-02 | 1.33e-02 | 3.13e-02 | 7.69e-02 | 6.72e-02 | 8.90e-02 | 9.68e+02 | 1.46e+02 |
| Xavier + BN | 1.14e-02 | 1.47e-02 | 2.68e-02 | 3.55e-02 | 8.25e-02 | 8.97e-02 | 4.50e-02 | 7.75e-01 |
| Xavier + LN | 2.36e-02 | 2.18e-02 | 3.07e-02 | 3.85e-01 | 1.40 | 4.69 | 2.60 | 6.14 |
| Proposed | **9.63e-03** | **8.29e-03** | **1.41e-02** | **1.88e-02** | **1.65e-02** | **1.85e-02** | **1.73e-02** | **3.59e-02** |

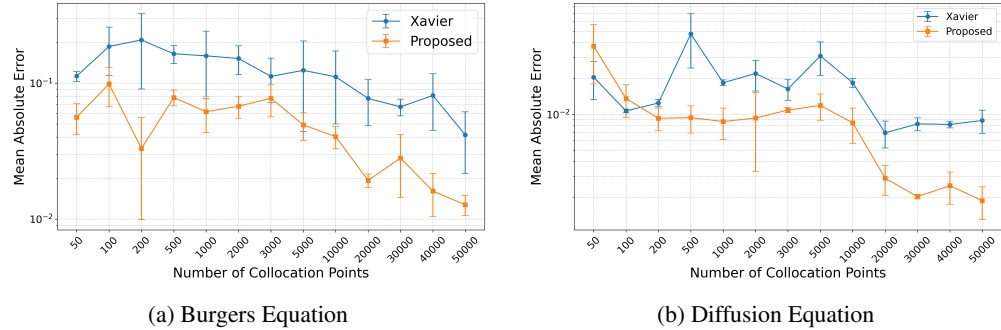

(a) Burgers Equation                          (b) Diffusion Equation

Figure 6: Mean absolute error between the exact solution and PINN-predicted solution with varying numbers of collocation points. The FFNN has 30 hidden layers (32 nodes each) and is trained for 300 iterations using Adam followed by 300 iterations using L-BFGS. The results are averaged over 5 experiments.

**Data Efficiency in PINNs.** Based on the results in Table 4, we evaluate data efficiency on a network with 30 hidden layers, each containing 32 nodes, where Xavier initialization achieved the lowest PINN loss. As shown in Figure 6, for the Burgers equation, the Mean Absolute Error (MAE) of the proposed initialization differs significantly from that of Xavier initialization across varying numbers of collocation points. In contrast, for the Diffusion equation, the difference in MAE between the two methods becomes more pronounced when the number of collocation points exceeds $20,000$. Additionally, Figure 7 illustrates that increasing the number of collocation points enables PINNs with the proposed initialization to predict solutions with lower absolute error. For detailed experiments on the Burgers equation, please refer to Appendix C.4.

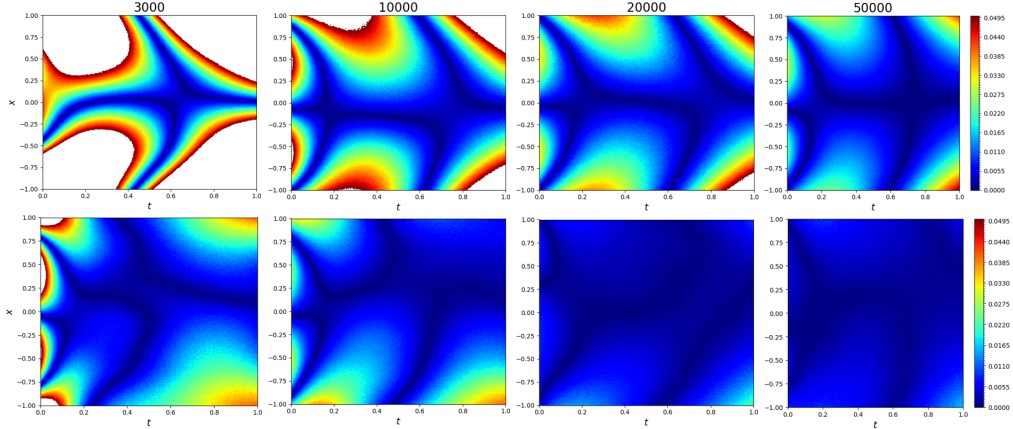

Figure 7: Absolute error between the exact solution and the PINN-predicted solution for the Diffusion equation with varying numbers of collocation points ($3000, 10000, 20000, 50000$) using **(upper row)** Xavier and **(lower row)** the proposed initialization. The FFNN has 30 hidden layers (32 nodes each) and is trained for 300 iterations using Adam followed by 300 iterations using L-BFGS. The color bar ranges from $0$ to $0.05$, with values outside this range shown in white.

## 5 CONCLUSION

In this study, we proposed a novel weight initialization method for tanh neural networks, baesed on a theoretical analysis of fixed points of $\tanh(ax)$ function. The proposed method is experimentally demonstrated to achieve robustness to variations in network size without normalization methods and to exhibit improved data efficiency. Therefore, the proposed weight initialization method reduces the time and effort required for training neural networks.

## ACKNOWLEDGMENTS

This work of Hyunwoo Lee and Hayoung Choi was supported by the National Research Foundation of Korea (NRF) grant funded by the Korea government (MSIT) (No. 2022R1A5A1033624 and RS-2024-00342939). The work of Hyunju Kim was supported by the funds of the Open R&D program of Korea Electric Power Corporation.(No-R23XO03).

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

# A    Analysis of Signal Propagation

In this section, we provide proofs for the statements presented in Section 3.1. Each proof follows from the fundamental properties of $\tanh(ax)$ and analytical derivations. An empirical analysis is performed on the activation distribution for normally and beta-distributed input data.

## A.1    Proof of Lemma 1

*Proof.* Define $g(x) = \tanh(ax) - x$. Since $g(x)$ is continuous, and $g(-M) > 0$, $g(M) < 0$ for a large real number $M$, the Intermediate Value Theorem guarantees the existence of a point $x$ such that $g(x) = 0$.

First, consider the case $0 < a \leq 1$. Since $0 < a \leq 1$, the derivative $g'(x) = a \cdot \text{sech}^2(ax) - 1$ satisfies $-1 \leq g'(x) \leq a - 1 < 0$ for all $x$. Hence, $g(x)$ is strictly decreasing and therefore $g(x)$ has a unique root. At $x = 0$, $\phi(0) = \tanh(a \cdot 0) = 0$. Hence, $x = 0$ is the unique fixed point.

We consider the case $a > 1$. For $0 < x \ll 1$, $\tanh(ax) - x \approx (a-1)x$. Since $a > 1$, $\tanh(ax) - x > 0$. On the other hand, since $|\tanh(ax)| < 1$ for all $x$,

$$\lim_{x \to \infty}[-1 - x] \leq \lim_{x \to \infty}[\tanh(ax) - x] \leq \lim_{x \to \infty}[1 - x].$$

By the squeeze theorem, $\lim_{x \to \infty}[\tanh(ax) - x] = -\infty$. By the intermediate value theorem, therefore, there exists at least one $x > 0$ such that $\tanh(ax) = x$. To establish the uniqueness of the positive fixed point, we investigate the derivative $g'(x) = a\,\text{sech}^2(ax) - 1$. We find the critical points to be $x = \pm\frac{1}{a}\sec^{-1}(\frac{1}{\sqrt{a}})$. It is straightforward to see that $g'(x) > 0$ in $\left(-\frac{1}{a}\sec^{-1}(\frac{1}{\sqrt{a}}), \frac{1}{a}\sec^{-1}(\frac{1}{\sqrt{a}})\right)$ and $g'(x) < 0$ in $\mathbb{R}\backslash\left(-\frac{1}{a}\sec^{-1}(\frac{1}{\sqrt{a}}), \frac{1}{a}\sec^{-1}(\frac{1}{\sqrt{a}})\right)$. i.e. $g(x) = 0$ has exactly two fixed points. Because $g(x)$ is an odd function, if $x^*$ is a solution, then $-x^*$ is also a solution. Thus, for $a > 1$, there exists a unique positive fixed point if $x > 0$ and a unique negative fixed point if $x < 0$. $\qquad\square$

## A.2    Proof of Lemma 2

*Proof.* (1) Since $(\tanh(ax))' = a\,\text{sech}^2(ax) < 1$ for all $x > 0$, it holds that $x_{n+1} = \phi_a(x_n) < x_n$ for all $n \in \mathbb{N}$. Thus the sequence $\{x_n\}_{n=1}^{\infty}$ is decreasing. Since $x_n > 0$ for all $n \in \mathbb{N}$, by the monotone convergence theorem, it converges to the fixed point $x^* = 0$.
(2) Let $x_0 < \xi_a$. Since $\phi'(x)$ is decreasing for $x \geq 0$, with $\phi'(0) > 1$ and $\xi_a$ is the unique fixed point for $x > 0$, it holds that $x_n < x_{n+1} < \xi_a$ for all $n \in \mathbb{N}$. Thus, by the monotone convergence theorem, the sequence converges to the fixed point $\xi_a$. The proof is similar when $x_0 > \xi_a$. By the monotone convergence theorem, the sequence also converges to the fixed point $\xi_a$. $\qquad\square$

## A.3    Proof of Proposition 3

*Proof.* Set $N = \max\{n | a_n > 1\}$. Define the sequences $\{b_n\}_{n=1}^{\infty}$ and $\{c_n\}_{n=1}^{\infty}$ such that $b_n = c_n = a_n$ for $n \leq N$, with $b_n = 0$ and $c_n = 1$ for $n > N$. Suppose that $\{\hat{\Phi}_m\}_{m=1}^{\infty}$ and $\{\tilde{\Phi}_m\}_{m=1}^{\infty}$ are sequences of functions defined as for each $m \in \mathbb{N}$

$$\hat{\Phi}_m = \phi_{b_m} \circ \phi_{b_{m-1}} \circ \cdots \circ \phi_{b_1}, \quad \tilde{\Phi}_m = \phi_{c_m} \circ \phi_{c_{m-1}} \circ \cdots \circ \phi_{c_1}.$$

Then the inequality $\hat{\Phi}_m \leq \Phi_m \leq \tilde{\Phi}_m$ holds for all $m$. By Lemma 1, for any $x \geq 0$, we have $\lim_{m \to \infty}\hat{\Phi}_m = 0$ and $\lim_{m \to \infty}\tilde{\Phi}_m = 0$. Therefore, the Squeeze Theorem guarantees that $\lim_{m \to \infty}\Phi_m(x) = 0$. $\qquad\square$

## A.4    Proof of Corollary 4

*Proof.* Set $N = \max\{n \mid a_n < 1 + \epsilon\}$. Define the sequence $\{b_n\}_{n=1}^{\infty}$ such that $b_n = a_n$ for $n \leq N$, and $b_n = 1 + \epsilon$ for $n > N$. The remainder of the proof is analogous to the proof of Proposition 3. $\qquad\square$

## A.5 Activation Distribution Based on Input Data Distribution

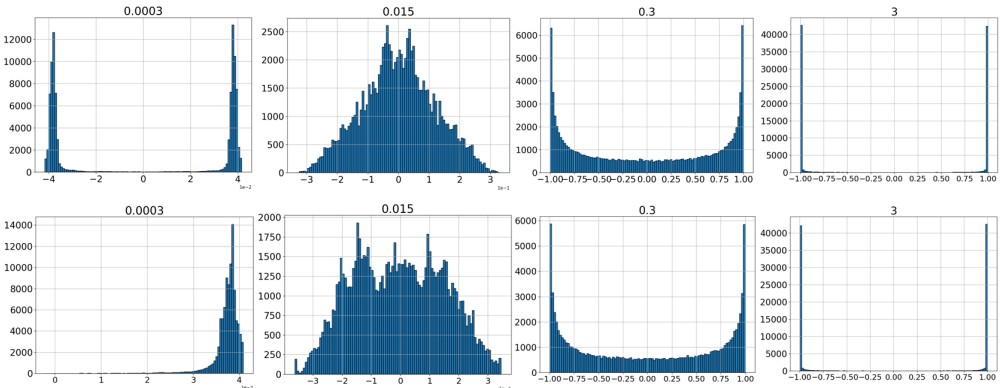

Figure 8: The activation values in the 1000th layer, with 32 nodes per hidden layer, were analyzed using the proposed weight initialization method with $\sigma_z$ values of 0.0003, 0.015, 0.3, and 3. The **upper row** shows results for 3000 input samples drawn from a standard normal distribution, while the **lower row** presents results for samples drawn from a Beta distribution with parameters $a = 2.0$ and $b = 5.0$.

## B Classification Tasks

In this section, we present experimental results for the benchmark classification datasets MNIST, Fashion-MNIST, CIFAR-10, and CIFAR-100 when employing the proposed weight initialization method.

### B.1 Width independence in Classification tasks

Here, we present the detailed experimental results of Table 1. Validation accuracy and loss for MNIST, Fashion-MNIST, CIFAR-10, and CIFAR-100 are presented for tanh FFNNs with varying numbers of nodes $(2, 8, 32, \text{and } 128)$, each with 20 hidden layers.

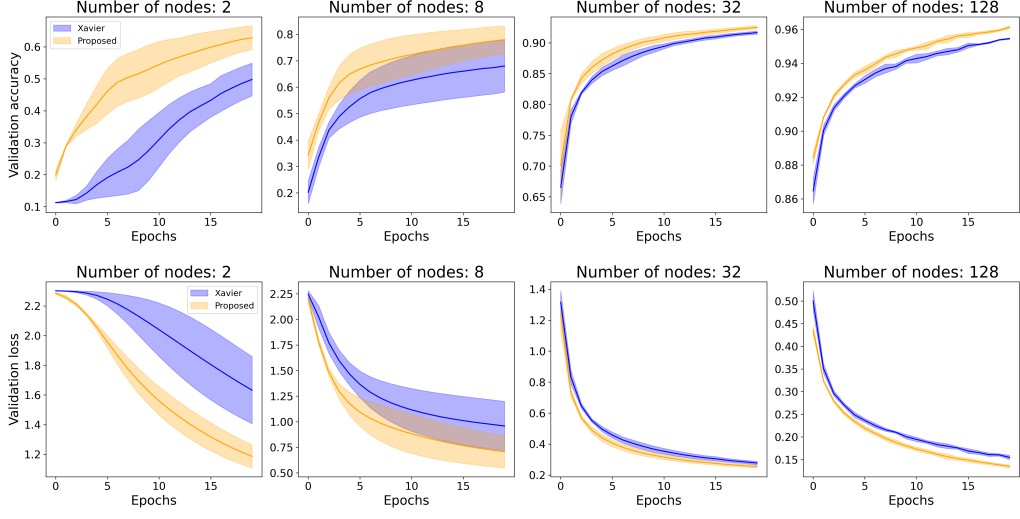

Figure 9: Validation accuracy and loss are presented for tanh FFNNs with varying numbers of nodes $(2, 8, 32, \text{and } 128)$, each with 20 hidden layers. All models were trained for 20 epochs on the MNIST dataset, with 10 different random seeds.

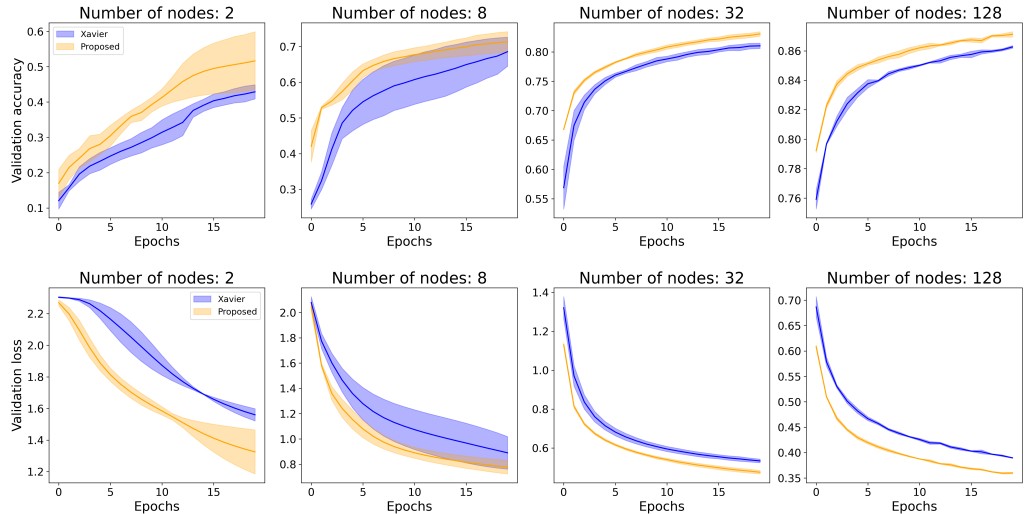

Figure 10: Validation accuracy and loss are presented for tanh FFNNs with varying numbers of nodes $(2, 8, 32, \text{and } 128)$, each with 20 hidden layers. All models were trained for 20 epochs on the Fashion MNIST dataset, with 10 different random seeds.

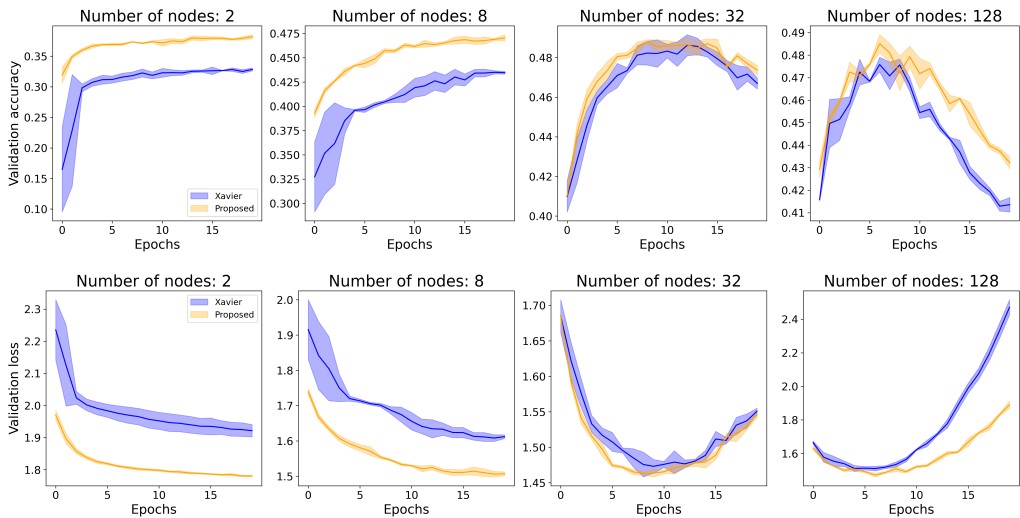

Figure 11: Validation accuracy and loss are presented for tanh FFNNs with varying numbers of nodes $(2, 8, 32, \text{and } 128)$, each with 20 hidden layers. All models were trained for 20 epochs on the CIFAR-10 dataset, with 10 different random seeds.

## B.2 NON-UNIFORM HIDDEN LAYER DIMENSIONS

Tanh neural networks have been less widely employed compared to ReLU networks due to their higher computational complexity, susceptibility to the vanishing gradient problem, and the superior empirical performance of ReLU in various deep learning tasks. However, the recent success of PINNs using tanh neural networks has renewed interest in their application. This section compares the performance of four initialization methods on architectures that are generally challenging to train: (1) tanh activation with Xavier initialization, (2) tanh activation with the proposed initialization, (3) ReLU activation with He initialization + BN, and (4) ReLU activation with orthogonal initialization. The experiments are performed on FFNN and autoencoder models.

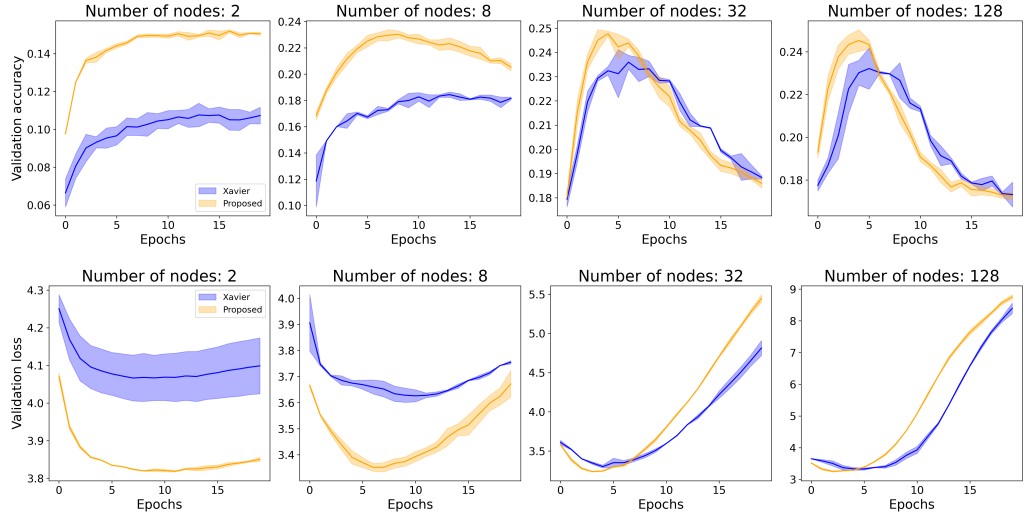

Figure 12: Validation accuracy and loss are presented for tanh FFNNs with varying numbers of nodes $(2, 8, 32, \text{and } 128)$, each with 20 hidden layers. All models were trained for 20 epochs on the CIFAR-100 dataset, with 10 different random seeds.

**FFNN.** The experiment is performed using an FFNN with a structure in which hidden layers alternated between 16 and 4 nodes, repeated 50 times, and trained over 100 epochs. The results are shown in Figure 13 (a). Both Xavier and the proposed method successfully train the network, with the proposed method showing overall better performance. Due to its strong performance despite substantial variations in hidden layer sizes, the proposed method is further evaluated on autoencoders with large variations in layer sizes, as shown in Figure 13 (b).

**Autoencoder.** The autoencoder architecture consists of an encoder and a decoder, both employing batch normalization and dropout (0.2) for regularization. The encoder compresses the input through layers of 512, 256, and 128 units before mapping to a 64-dimensional latent space. The decoder reconstructs the input by symmetrically expanding the latent space through layers of 128, 256, and 512 units, followed by an output layer with sigmoid activation. In Figure 13 (b), the model is trained on the MNIST dataset with a batch size of 256, while in (c), it is trained on the FMNIST dataset with a batch size of 512.

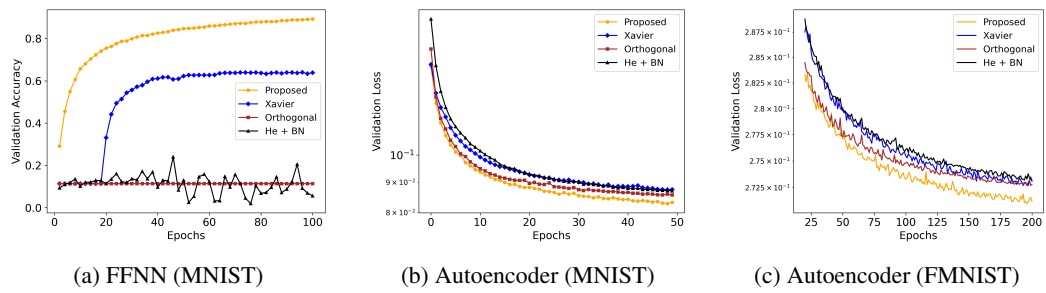

    (a) FFNN (MNIST)          (b) Autoencoder (MNIST)        (c) Autoencoder (FMNIST)

Figure 13: **(a)** Validation loss for an FFNN with alternating hidden layers of 16 and 4 nodes, repeated 50 times, comparing four methods: Tanh with Xavier initialization, Tanh with the proposed initialization, ReLU with He initialization + BN, and ReLU with orthogonal initialization. **(b)** Validation loss for an autoencoder with encoder-decoder layers of 512, 256, 128, and 64 units, comparing the same four methods. **(c)** Same as (b), but on the FMNIST dataset.

## C  PHYSICS-INFORMED NEURAL NETWORKS

In this section, we present additional experiments employing the proposed weight initialization method in Physics-Informed Neural Networks (PINNs). Specifically, we empirically analyze its relationship with different activation functions (Swish, ELU, Sigmoid, and ReLU) and its impact on various PDE problems (Burgers' equation, Allen-Cahn equation, Poisson equation, and Diffusion equation).

### C.1  PDE DETAILS

Here we present the differential equations that we study in experiments.

**Allen-Cahn Equation.** The diffusion coefficient is set to $d = 0.01$. The initial condition is defined as $u(x, 0) = x^2 \cos(\pi x)$ for $x \in [-1, 1]$, with boundary conditions $u(-1, t) = -1$ and $u(1, t) = -1$, applied over the time interval $t \in [0, 1]$. The Allen-Cahn equation is expressed as

$$\frac{\partial u}{\partial t} - d \frac{\partial^2 u}{\partial x^2} = -\frac{u^3 + u}{d},$$

where $u(x, t)$ represents the solution, $d$ is the diffusion coefficient, and the nonlinear term $u^3 - u$ models the phase separation dynamics.

**Burgers' Equation.** The Burgers' equation, a viscosity coefficient of $\nu = 0.01$ is employed. The initial condition is given by $u(x, 0) = -\sin(\pi x)$ for $x \in [-1, 1]$, with boundary conditions $u(-1, t) = 0$ and $u(1, t) = 0$ imposed for $t \in [0, 1]$. The Burgers' equation is expressed as

$$\frac{\partial u}{\partial t} + u \frac{\partial u}{\partial x} = \nu \frac{\partial^2 u}{\partial x^2},$$

where $u(x, t)$ is the velocity field, and $\nu$ is the viscosity coefficient.

**Diffusion Equation.** The diffusion equation includes a time-dependent source term and is defined over the spatial domain $x \in [-1, 1]$ and temporal interval $t \in [0, 1]$. The initial condition is specified as $u(x, 0) = \sin(\pi x)$, with Dirichlet boundary conditions $u(-1, t) = 0$ and $u(1, t) = 0$. The diffusion equation is expressed as

$$\frac{\partial u}{\partial t} - \frac{\partial^2 u}{\partial x^2} = e^{-t} \left( \sin(\pi x) - \pi^2 \sin(\pi x) \right),$$

where $u(x, t)$ is the solution.

**Poisson Equation.** The Poisson equation is defined over the spatial domain $x \in [0, 1]$ and $y \in [0, 1]$. The Poisson equation is expressed as

$$\frac{\partial^2 u}{\partial x^2} + \frac{\partial^2 u}{\partial y^2} = f(x, y),$$

where $u(x, y)$ is the solution, and $f(x, y)$ is the source term given by

$$f(x, y) = 2\pi^2 \sin(\pi x) \sin(\pi y).$$

### C.2  EFFECT OF ACTIVATION FUNCTION ON PINNs

We experimentally demonstrate that when employing the proposed weight initialization method, the absolute error between the exact solution and the PINN-predicted solution is smaller when using the tanh activation compared to ReLU, sigmoid, Swish (Ramachandran et al., 2017), and ELU (Clevert, 2015), as shown in Figure 14. The FFNN consists of 30 hidden layers (32 nodes each) and is trained for 300 iterations using Adam, followed by 300 iterations using L-BFGS. For the Burgers' and diffusion equations, the PINN with tanh activation achieves the closest approximation to the exact solution. It is well established that PINNs with tanh activation effectively approximate

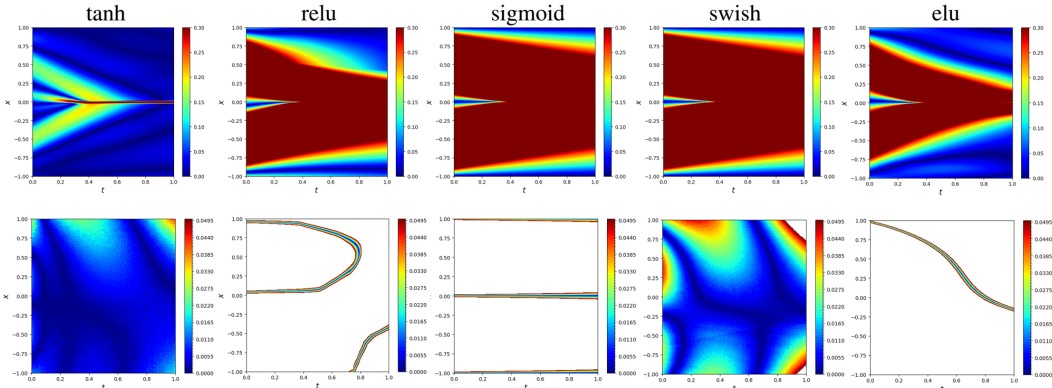

Figure 14: Absolute error for **(upper row)** the Burgers' equation and **(lower row)** the diffusion equation with varying activation functions. Values outside the color bar range are depicted in white.

solutions to the Burgers' and diffusion equations. Given that the choice of activation function and weight initialization are dependent, this result provides valuable insights into the interaction between initialization methods and activation functions in PINNs..

Figure 15 presents experiments that employ the proposed initialization and Xavier initialization for PDE problems using the Swish activation function, which is widely used in PINNs.

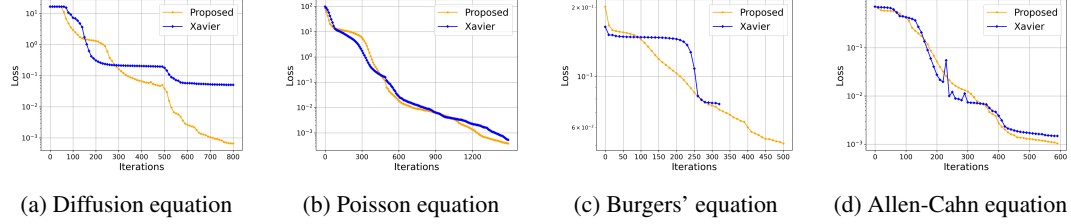

(a) Diffusion equation      (b) Poisson equation      (c) Burgers' equation      (d) Allen-Cahn equation

Figure 15: PINN loss for a Swish FFNN with (a) 20 hidden layers, each containing 32 nodes, and (b), (c) 3 hidden layers, each containing 3 nodes, and (d) 10 hidden layers, each containing 32 nodes.

## C.3    $\sigma_z$ FOR BURGERS' EQUATION

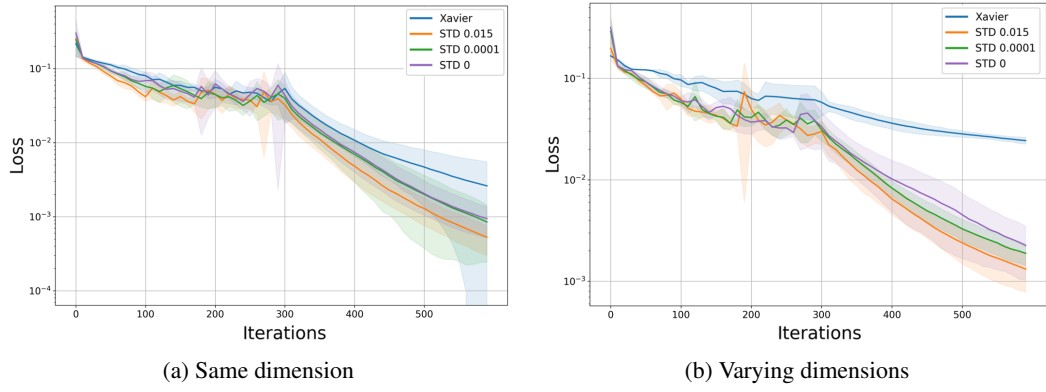

(a) Same dimension                (b) Varying dimensions

Figure 16: Here, STD refers to $\sigma_z$. (a) shows the PINN loss for the Burgers' equation, using an FFNN with 30 layers and 32 nodes in each hidden layer. (b) shows the PINN loss for an FFNN with 30 layers, where the hidden layers alternate between 64 and 32 nodes, repeated 15 times.

Here, we present the PINN loss for the Burgers' equation as a function of the initialization hyper-parameter $\sigma_z$. As shown in Figure 16, the experimental results demonstrate that $\sigma_z$ significantly impacts training. Therefore, selecting an appropriate $\sigma_z$ is crucial for optimizing performance.

### C.4 ABSOLUTE ERROR FOR BURGERS' EQUATION

Figure 17 demonstrates that increasing the number of collocation points allows PINNs with the proposed initialization to predict solutions with lower absolute error compared to those using Xavier initialization. The proposed method is experimentally demonstrated to achieve faster convergence speed and higher data efficiency.

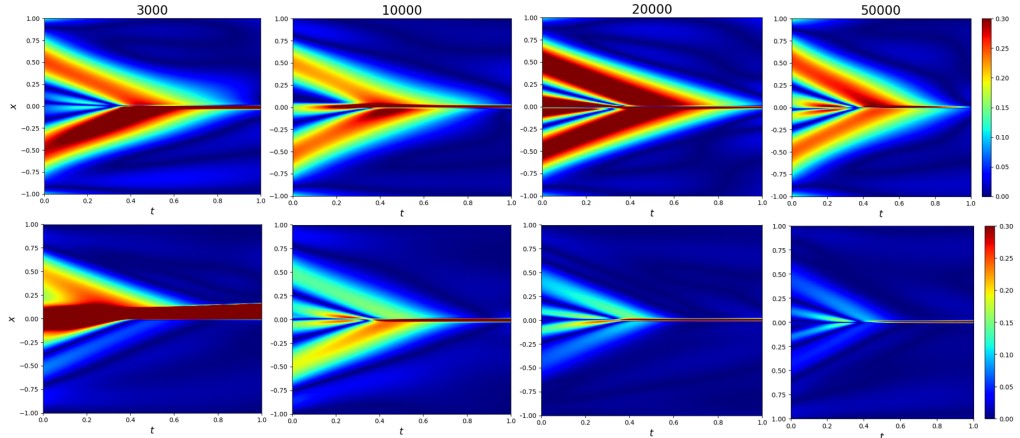

Figure 17: Absolute error between the exact solution and the PINN-predicted solution for the Burgers' equation with varying numbers of collocation points (3000, 10000, 20000, 50000) using **(upper row)** Xavier and **(lower row)** the proposed initialization. The FFNN has 30 hidden layers (32 nodes each) and is trained for 300 iterations using Adam followed by 300 iterations using L-BFGS.

## D  EXAMPLES OF THE MATRIX $\mathbf{D}^\ell$

An example of $\mathbf{D}^\ell$ from the initialization methodology proposed in Section 3.2 is presented. In the heatmap of Figure 18, white represents the value 0, while black represents the value 1.

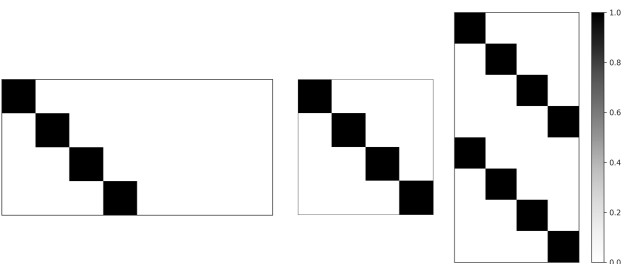

Figure 18: Examples of the matrix $\mathbf{D}^\ell \in \mathbb{R}^{N_\ell \times N_{\ell-1}}$ in Section 3.2 with $N_\ell = 4, N_{\ell-1} = 8$ (left), $N_\ell = 4, N_{\ell-1} = 4$ (middle), and $N_\ell = 8, N_{\ell-1} = 4$ (right), respectively.

