# OpenReview forum: "Robust Weight Initialization for Tanh Neural Networks with Fixed Point Analysis"
_ICLR.cc/2025/Conference — ICLR 2025 Poster_

### Official Review · Reviewer_YCAx · 2024-10-25

**Soundness:** 3
**Presentation:** 3
**Contribution:** 3
**Rating:** 6
**Confidence:** 4

**Summary:**

The paper proposes a new method to initialize weights for FCNNs and PINNs with tanh activation. The paper claims that the proposed weight initialization method will not lead to diminishing activations for very deep networks unlike Xavier weight initialization. The paper claims that the proposed weight initialization is robust to network depth and number of units in hidden leayers.

**Strengths:**

Originality
The paper presents a novel weight initialization method specifically designed for tanh-based neural networks, addressing an understudied area in neural network initialization. This approach is distinct in its use of fixed-point analysis to prevent activation saturation and improve training robustness across network sizes, particularly in the context of Physics-Informed Neural Networks (PINNs). By emphasizing robustness and performance consistency in both traditional classification tasks and PINNs, the paper makes a valuable contribution to the field of weight initialization. The originality is strong, given the lack of prior research focusing on tanh-based initialization methods.

Quality
The paper demonstrates high quality in both theoretical and experimental aspects. The method is grounded in rigorous mathematical analysis, leveraging fixed-point theory to derive conditions that ensure stable activation propagation. The provided lemmas, proofs, and propositions add credibility and depth to the approach.
Experiments are well-designed and span various network configurations, datasets (MNIST, Fashion MNIST, CIFAR-10), and applications (PINNs for solving differential equations). The results consistently show that the proposed method outperforms Xavier initialization, particularly in deeper networks and varying network sizes.

Clarity:
Overall the paper is very well written, with some exceptions mentioned in the weakness section. All the sections in the paper are laid out clearly. The notations are consistent across the paper.

**Weaknesses:**

Significance:
The paper compares their proposed weight initialization with Xavier weight initialization for FCNN with tanh activation. Xavier is known to show diminishing gradients and activations problem for deeper networks, but this is solved by using layer normalization. I am therefore considering this work not significant because the problem that the authors are trying to solve does not exist for Xavier + Layer norm and the authors did not do any comparative analysis with and without layer norm.

Other major issue:
1. In equation 2, the paper claims that a_i^(k+1) follows normal distribution with unit mean. But when number of neurons in layer l-1 (N_(l-1)) is greater than number of neurons in layer l (N_l), then the mean will be greater than 1. When N_(l-1) = 2 * N_l, the mean will be 2. This is not clearly discussed in the paper. If the mean is > 1, then that leads to tanh always saturated.

**Questions:**

1. Layer norm is added to handle the diminishing activation problem. Any reason why you did not compare the performance of the proposed approach with Xavier weight initialization with Layer norm?

2. In equation 2, the paper claims that a_i^(k+1) follows normal distribution with unit mean. But when number of neurons in layer l-1 (N_(l-1)) is greater than number of neurons in layer l (N_l), then the mean can be greater than 1. When N_(l-1) = 2 * N_l, the mean will be 2. This is not clearly discussed in the paper. If the mean is > 1, then that can lead to tanh saturation.

---

> ### Author Response · Authors · 2024-11-24
>
> Thank you for your valuable comments! In the following section, we will address the weaknesses (W) and questions (Q) mentioned above. **The changes are marked in blue.**
>
> **W1&Q2** *"In equation 2, the paper claims that $a_i^{k+1}$ follows normal distribution with unit mean. But when number of neurons in layer l-1 (N_(l-1)) is greater than number of neurons in layer l (N_l), then the mean will be greater than 1. When N_(l-1) = 2 * N_l, the mean will be 2. This is not clearly discussed in the paper. If the mean is > 1, then that leads to tanh always saturated."*:
>
> A1: Thank you for your valuable comment. We propose a weight matrix defined as $\mathbf{W}^{\ell} = \mathbf{D}^{\ell} + \mathbf{Z}^{\ell} \in \mathbb{R}^{N_{\ell} \times N_{\ell-1}}$, where $\mathbf{D_{i,j}}^{\ell} = 1$ if $i \equiv j \pmod{N_{\ell-1}}$, and 0 otherwise.  If $\mathbf{D_{i,j}}^{\ell} = 1$ when $i \equiv j \pmod{N_{\ell}}$, and 0 otherwise, the mean in the mentioned case could become 2. However, in the proposed method, all off-diagonal elements of $\mathbf{D}^{\ell}$ are 0 when $N_{\ell-1} > N_{\ell}$, ensuring that the mean remains 1. We have revised the manuscript as follows to improve clarity:
> - **Inline Headings were added** to each paragraph for better organization.
> - In Section 3.2, we included a **remark outlining two conditions** that the proposed weight initialization is designed to satisfy.
>
> **Q1** *"Layer norm is added to handle the diminishing activation problem. Any reason why you did not compare the performance of the proposed approach with Xavier weight initialization with Layer norm?"*:
>
> A2: Thank you for your valuable suggestion. To the best of our knowledge, in neural networks using the tanh activation function, normalization methods are less effective due to the inherent properties of tanh. Tanh suffers from gradient saturation for large inputs, where gradients approach zero, and normalization cannot effectively mitigate this issue. Additionally, normalization methods generally introduce a computational overhead of approximately 30% and require additional time for tuning, such as deciding how frequently to apply them across layers. In response to the reviewers' comments, we conducted experiments to compare the proposed method with Xavier initialization, both with and without normalization methods (Batch Normalization and Layer Normalization), and revised the paper as follows:
>
> - **In Figure 4**, we validated Xavier, Xavier with BN, Xavier with LN, and the proposed method on MNIST, FMNIST, CIFAR-10, and CIFAR-100 datasets. The proposed method demonstrated the fastest convergence and the highest accuracy across all datasets.
> - **In Table 3**, we evaluated the **data efficiency** of the four methods on MNIST and FMNIST. The results show that the proposed method achieved higher accuracy even with limited data.
> - **In Table 4**, we tested the four methods on **the Allen-Cahn, Burgers, Diffusion, and Poisson equations** across various network sizes. The proposed method exhibited the greatest robustness to network size.
>
> We hope that these explanations address your concerns, but we'd be happy to answer any remaining questions about our method.

---

> ### Author Response · Authors · 2024-11-25
> **Reminder for Review YCAx**
>
> Dear Reviewer YCAx,
>
> We hope that our responses could adequately address your concerns. As the discussion phase deadline approaches, we warmly welcome further discussion regarding any additional concerns that you may have, and we sincerely hope you can reconsider the rating accordingly.
>
> Thank you for the time and appreciation that you have dedicated to our work.
>
> Best regards,
>
> Authors of submission 13420

---

> ### Comment · Reviewer_YCAx · 2024-11-26
> **Response to Author**
>
> Thank you for addressing the comments and questions.
>
> **Regarding W1&Q2**
> * I don't see where this is mentioned in the paper "all off-diagonal elements of $D_{i,j}^l$ are 0, ensuring that the mean remains 1"
> * Also please clarify what do you mean by diagonal in a rectangular matrix
> * What about the case where $N_{l-1} < N_l$, the mean will be less than 1 even when off diagonal elements are zero.
>
> **Regarding Q1** \
> Thank you for adding the results with layer norm and batch norm.

---

> ### Author Response · Authors · 2024-11-26
>
> **Regarding W1&Q2**
>
> **1.** *"I don't see where this is mentioned in the paper "all off-diagonal elements of $D_{i,j}^l$ are 0, ensuring that the mean remains 1."*:
>
> A1: Thank you for your comment. In the revised paper (lines 255--256), the proposed weight initialization method defines $D_{i,j}^{\ell} = 1$ if $i \equiv j \pmod{N_{\ell -1}}$, and 0 otherwise. From this definition of $D^{\ell}$, it can be inferred that when $N_{\ell-1} > N_{\ell}$, all off-diagonal elements of $D^{\ell}$ are 0.
>
> To clarify, the condition $i \equiv j \pmod{N_{\ell-1}}$ implies that $j = i + k \cdot N_{\ell-1}$, where $k \in \mathbb{Z}$. Additionally, $j$ must lie within the valid range of indices $1 \leq j \leq N_{\ell-1}$. This restricts the possible values of $j$ for a given $i$. When $N_{\ell-1} > N_{\ell}$, the only valid $j$ that satisfies $i \equiv j \pmod{N_{\ell-1}}$ and lies within the range $1 \leq j \leq N_{\ell-1}$ is $j = i$. All other values of $j \neq i$ fall outside the valid range due to the modular condition and the size constraints of the indices. Therefore, under the condition $N_{\ell-1} > N_{\ell}$, all off-diagonal elements of $\mathbf{D}^{\ell}$ are zero.  In response to the reviewers' comments, we have revised the manuscript for clarification, as follows, instead of providing a formal proof.
>
> - We have **added examples** of $D^{\ell}$ for the cases $N_{\ell} < N_{\ell-1}$, $N_{\ell} = N_{\ell-1}$, and $N_{\ell} > N_{\ell-1}$ in **Figure 17**.
>
> **2.** *"Also please clarify what do you mean by diagonal in a rectangular matrix."*:
>
> A2: We apologize for any confusion caused by our previous response. In a rectangular matrix, the diagonal specifically refers to the elements where the row and column indices are equal.
>
> **3.** *"What about the case where $N_{l-1}<N_{l}$, the mean will be less than 1 even when off diagonal elements are zero.
> "*:
>
> A3: Thank you for your valuable comment. To address the case where $N_{\ell-1} < N_{\ell}$, let us provide an example. Assume $D^{\ell}$ is a $4 \times 2$ matrix, and $Z^{\ell}$ is added such that all off-diagonal elements remain 0. The resulting $W^{\ell}$ is as follows:
>
> [[1+z_1,  0],
>
> [0,   1+z_2],
>
> [1+z_3,   0 ],
>
> [0,   1+z_4]]
>
> where $z_1, \dots, z_4$ are drawn from a normal distribution with mean 0 and variance $\sigma_z^2$ by definition of prposed method. For an input vector $x^0 = (x^0_1, x^0_2)$, the output becomes $x^1_1 = \tanh((1+z_1) \cdot x^0_1)$. Therefore, even in this case, the mean remains 1 because the added noise $z_i$ has a mean of 0. This example is applicable to matrices when $N_{l-1}> N_l$. We have revised the manuscript for clarification as follows:
>
> - We have **added examples** of $D^{\ell}$ for the cases $N_{\ell} < N_{\ell-1}$, $N_{\ell} = N_{\ell-1}$, and $N_{\ell} > N_{\ell-1}$ in **Figure 17**.
>
> **Regarding Normalization**
>
> We sincerely appreciate your valuable suggestions regarding normalization, which have significantly improved our manuscript. We hope the **experiments** provided address your concerns effectively.
>
> We warmly welcome any further discussion or additional feedback you may have and kindly hope you might reconsider your evaluation in light of these improvements. Thank you again for your thoughtful review.

---

### Official Review · Reviewer_kh6z · 2024-10-25

**Soundness:** 2
**Presentation:** 3
**Contribution:** 2
**Rating:** 5
**Confidence:** 3

**Summary:**

Weight initialisation is an old topic, and most studies have verified that weight initialization methods can improve the performance of neural networks. However, because that neural network’s depth increasing rapidly,  most neural networks, especially Feedforward Neural Networks (FFNNs) should face the gradient vanishment problem.  In this article, authors proposed one weight initialisation method for FFNNs and Physics-Informed Neural Networks (PINNs). Based on an analysis of the fixed points of the function tanh(ax), this method determines values of $a$ that prevent the saturation of activations during the training progress. In terms of robustness, this method presents a stronger and more efficient performance. In the experiment, verified on MNIST, Fashion MNIST, and CIFAR10 datasets, this method also shows acceptable results compared to the Xavier.

**Strengths:**

1. this article proposes one novel weight initialization method for the FFNN and PINN.
2. the authors prove that the activation values cannot vanish when increasing the depth of the neural network by using a fixed-point analysis.

**Weaknesses:**

1. The universality of this method should be improved. For example, more experiments on the feasibility of other neural networks except FNN and PINN.
2. The details of hyperparameters should be mentioned, such as what is the $Threshold$ of FFNN, which training strategy (supervised or unsupervised) of FFNN is used?
3. FFNN is specifically designed to visualize the trained features, which also should be discussed.
4. The consistency of results should be guaranteed. For example, in Figure 3 (c) and (d), there is a different trend (Xavier tends to equal
the proposed method), which also should be discussed. In case that after 20 epochs, the performance would be totally different to the presented result.

**Questions:**

The novelty of this work is strong, and the topic sounds interesting. However, the writing and structure should be revised again. There are some questions that the authors should be concerned about.

1. In Eq. 2, $\sigma_{z}$ is set to $\alpha/\sqrt{N^{l}-1}$ and $\alpha = 0.085$. From Figure 2, we can find that the optimal value of $\alpha$ is $0.085$. Is there any theoretical reason why $\alpha$ should be $0.085$. Or should we manually try the value accordingly?

2. Additionally, please scribe what is $\alpha$. There is no definition of $\alpha$.

3. In Figure 3 (c) and (d), the proposed method seems to decrease after 6 epochs. Although the accuracy curve can rapidly reach the peak (faster than Xavier), the robustness of this method also should be discussed. For example, the initialization method can first provide prior knowledge to neural networks, but if it can keep the stability of training or not. Or is it the reason for the high $\alpha$?

4. In Appendix A.1, the authors discussed different conditions. When $x = 0$, whether the vanishment problem will abscond. Please highlight the strategy on how this method can process it.

---

> ### Author Response · Authors · 2024-11-24
>
> Thank you for your valuable comments! In the following section, we will address the weaknesses (W) and questions (Q) mentioned above. **The changes are marked in blue.**
>
> **W1** *"The universality of this method should be improved. For example, more experiments on the feasibility of other neural networks except FNN and PINN."*:
>
> A1: Thank you for your valuable suggestion. We derived the initialization by simplifying the process of signal propagation in feedforward neural networks, making it particularly effective for architectures using tanh FFNNs. One such example is Physics-Informed Neural Networks (PINNs). In response to the reviewers' comments, we have added new experiments as follows:
> - We have added **experiments on autoencoders in Figure 13 (b) and (c)** to compare the performance of four methods: (1) tanh activation with Xavier initialization, (2) tanh activation with the proposed initialization, (3) ReLU activation with He initialization and Batch Normalization (BN), and (4) ReLU activation with orthogonal initialization. The results show that the proposed method achieves the fastest convergence and the lowest validation loss among all methods.
>
> **W2** *"The details of hyperparameters should be mentioned, such as what is the threshold  of FFNN, which training strategy (supervised or unsupervised) of FFNN is used?"*:
>
> A2: Thank you for your comment. In response to the reviewers' comments, we have revised the manuscript as follows:
> - We have specified the experimental settings as inline headings in Section 4 and revised them in enhanced detail.
>
> **W3** *"FFNN is specifically designed to visualize the trained features, which also should be discussed."*
>
> A3: Thank you for your comment. We observed changes in the weight matrix during the training of the FFNN. Additionally, we analyzed the changes in the rank, eigenvalues and eigenvectors of this matrix, which provided valuable insights into the training process of the FFNN. However, as the primary focus of this paper is on effectively propagating signals through deep layers, we chose not to visualize the weight matrix to maintain the coherence of the manuscript.
>
> **W4** *"The consistency of results should be guaranteed. For example, in Figure 3 (c) and (d), there is a different trend (Xavier tends to equal the proposed method), which also should be discussed. In case that after 20 epochs, the performance would be totally different to the presented result."*:
>
> A4:  Thank you for your valuable suggestion. In response to the reviewers' comments, we have added new experiments:
> - We have added experiments with **100 epochs of training on MNIST, FMNIST, CIFAR-10, and CIFAR-100 in Figure 4**.
>  The proposed method achieved the highest accuracy and the fastest convergence across all four datasets.
>
> **W5** *" The writing and structure should be revised again"*:
>
> A5: Thank you for your comment. In response to the reviewers' comments, we have revised the manuscript as follows:
> - We have made minor edits for **clarity** and **space efficiency**.
> - We have added **inline headings** to each paragraph for improved organization.
> - We have added **explanations in Section 3.1** to enhance understanding.
> - We have specified the conditions that the initial weight matrix should satisfy in the **remark of Section 3.2**.

---

> ### Author Response · Authors · 2024-11-25
>
> **Q1&Q2** *"What is  𝛼 "*:
>
> A6: We propose a weight initialization method that satisfies the following condition during the initial forward pass: it ensures that the distribution of activation values in deeper layers is approximately normal. As shown in Figures 3 and 8, when $\sigma_z = 0.015$, the activation value distribution in the 1000th layer is observed to be approximately normal distribution. In this experiment, all hidden layers were set to have 32 nodes, satisfying  $0.015 = \alpha / \sqrt{32}$. The value of 𝛼 is approximately 0.085. In response to the reviewers' comments, we have revised the manuscript.
>
> - We have added additional **details about 𝛼 in the final paragraph of Section 3.3**.
> - We have specified the conditions that the initial weight matrix should satisfy in the **remark of Section 3.2**.
> - We have added **experiments on the changes in activation distribution with respect to $\sigma_z$ in Figures 3 and 8**.
>
>
> **Q3** *"In Figure 3 (c) and (d), the proposed method seems to decrease after 6 epochs. Although the accuracy curve can rapidly reach the peak (faster than Xavier), the robustness of this method also should be discussed. For example, the initialization method can first provide prior knowledge to neural networks, but if it can keep the stability of training or not. Or is it the reason for the high 𝛼? "*:
>
> A7: Thank you for this valuable suggestion. In response to the reviewers' comments, additional experiments were conducted:
> - We **trained models on the MNIST, FMNIST, CIFAR-10, and CIFAR-100 datasets for up to 100 epochs in Figure 4**. For the MNIST and FMNIST datasets, all four initialization methods converged after 40 epochs. However, for CIFAR-10 and CIFAR-100, all four initialization methods reached a peak in accuracy at around 10 epochs, followed by a rapid decline. The proposed method, as shown in Figures 4 (c) and (d), reached the peak faster and exhibited a slower decline in accuracy compared to the other methods. These results demonstrate that the proposed method achieves both rapid convergence and improved training stability compared to existing methods.
> - The proposed value of $\alpha=0.085$, as shown in **Figure 3 and 8, prevents activation values from saturating even at the 1000th layer** and maintains a consistent scale for activation values, as illustrated in Figure 1. This ensures that signals are not lost in deeper layers.
>
>
>
>
> **Q4** "In Appendix A.1, the authors discussed different conditions. When x= 0, whether the vanishment problem will abscond. Please highlight the strategy on how this method can process it.":
>
> A8: Thank you for pointing out the need for further clarification of the strategy in Appendix A.1. In Appendix A.1, the proof approaches two cases:
> 1. When $\alpha \leq 1$, the fixed point $x^*=0$ is unique, and the activation values tend to shrink toward 0 as the network depth increases. This highlights the challenge of the vanishing activation problem.
> 2. When $\alpha > 1$, two fixed points $x^*=\pm\xi_a$ emerge, which prevent the activations from collapsing to zero.
>
> The key strategy of our method lies in ensuring that the initialization keeps $a$ close to 1 across the network, thus avoiding the vanishment of activations. If $x_i=0$ at initialization and all other elements of vector $x$ are non-zero, the proposed weight initialization ensures that subsequent activations will move away from zero. In response to the reviewers' comments, we have revised the manuscript as follows.
> - We have added **additional explanations in Section 3.1** for further clarification.
>
> We hope that these explanations address your concerns, but we'd be happy to answer any remaining questions about our method.

---

> ### Author Response · Authors · 2024-11-25
> **Reminder for Review  kh6z**
>
> Dear Reviewer kh6z,
>
> We hope that our responses could adequately address your concerns. As the discussion phase deadline approaches, we warmly welcome further discussion regarding any additional concerns that you may have, and we sincerely hope you can reconsider the rating accordingly.
>
> Thank you for the time and appreciation that you have dedicated to our work.
>
> Best regards,
>
> Authors of submission 13420

---

> ### Author Response · Authors · 2024-11-29
> **Kind Reminder for Reviewer kh6z**
>
> Dear reviewer  kh6z
>
> The extended discussion closes in few days.
> We've tried to address all your concerns with new results,
> clarifications and an updated manuscript.
> Please let us know if you have any remaining concerns.
> We look forward to a productive discussion, and we sincerely hope you can reconsider the rating accordingly.
>
> Best regards,
>
> Authors of submission 13420

---

> ### Author Response · Authors · 2024-12-03
> **[Reminder] Could you kindly verify if the provided clarification addresses your concerns?**
>
> Dear Reviewer kh6z,
>
> We believe we have carefully and comprehensively addressed all your concerns and questions.
> As the discussion period is set to close in less than 7 hours, we would be grateful for any additional suggestions or specific points to further enhance our manuscript.
>
> We sincerely hope you might reconsider your score or share further insights to help us strengthen our work.
>
> Thank you once again for your support and thoughtful guidance throughout this process.
> We deeply value your time and effort in reviewing our work.
>
> Best regards,
>
> Authors of submission 13420

---

### Official Review · Reviewer_6yoV · 2024-10-29

**Soundness:** 3
**Presentation:** 3
**Contribution:** 2
**Rating:** 6
**Confidence:** 2

**Summary:**

This work first provides a theoretical analysis of weight initialization when exclusively using tanh
as an activation function. Providing reasons for the clustering behavior when initializing networks.
Based on the developed theory, an initialization scheme is proposed, and the hyperparameter σz
is empirically determined. There are two types of experiments mainly comparing the proposed
initialization with Xavier. The first type of experiment concerns the classification accuracy in the
early training phase. Showing there is an improvement in accuracy across different data sets and
configurations. The second type of experiment concerns solving PDE with PINNS, showing that the
proposed method has a good performance.

**Strengths:**

Unlike optimization problems with theoretical guarantees on fixed points, weight initialization
is an important task in deep learning. An initialization scheme with theoretical backing can
have a long-lasting impact, even just for a sub-field of deep learning.

Experiments do show significant improvement when using PINNs to solve PDE.

**Weaknesses:**

The impact depended on exclusively using tanh as an activation function is fundamentally
beneficial in PINNs. As the current state of the paper, there is not enough support for this.

Given that the experiments are not too computationally intensive and the experiment section
only considers a few data sets or PDEs, the demonstrated improvement may not be general.

**Questions:**

In sections 4.1 and 4.2, the experiment trains for 20 or 40 epochs. Do networks converge to their best accuracy? What is the difference in accuracy when training for more epochs?

The experiments in section 4 are not too computationally intensive, is it possible to include more
data sets or PDE can show the improvements are general?

Can the code used in the experiment can be provided to improve reproducibility?

---

> ### Author Response · Authors · 2024-11-23
>
> Thank you for your valuable comments! In the following section, we will address the weaknesses (W) and questions (Q) mentioned above. **The changes are marked in blue.**
>
> **W1** *"The impact depended on exclusively using tanh as an activation function is fundamentally beneficial in PINNs. As the current state of the paper, there is not enough support for this."*:
>
> A1: Thank you for your insightful suggestion. The tanh activation function has been experimentally shown to perform well in PINNs, which is why tanh neural networks remain widely used [1,2,3,4]. In response to the reviewers' comments, we have revised the manuscript as follows.
> - We have presented the absolute error between the exact solution and the PINN-predicted solution for **different activation functions in Figure 14**. Tanh, sigmoid, swish, and ReLU activation functions were compared, with tanh showing the lowest absolute error.
>
> **Q1** *"In sections 4.1 and 4.2, the experiment trains for 20 or 40 epochs. Do networks converge to their best accuracy? What is the difference in accuracy when training for more epochs?"*:
>
> A2: Thank you for your valuable suggestion. In response to the reviewer’s comment, we conducted the following additional experiment:
> - We trained models on **MNIST, FMNIST, CIFAR-10, and CIFAR-100 datasets for up to 100 epochs in Figure 4**. For the MNIST and FMNIST datasets, all four initialization methods showed convergence after 40 epochs.  Notably, Xavier with normalization showed faster convergence compared to Xavier without normalization. In contrast, for CIFAR-10 and CIFAR-100, all methods exhibited overfitting tendencies. Across all four datasets, the proposed method achieved the highest accuracy.
>
> **W2&Q2** *"Experiment section only considers a few data sets or PDEs"*:
>
> A3: Thank you for your valuable suggestion. In response to the reviewers' comments, additional experiments were conducted, and the manuscript has been revised as follows.
> - Normalization methods were proposed to address gradient issues in deep neural networks. Therefore, we conducted experiments by applying Batch Normalization or Layer Normalization to Xavier initialization, as shown in **Tables 3 and 4 and Figure 4**. The results demonstrate that the proposed method, without normalization, achieves faster convergence, improved data efficiency, and is more robust to variations in network size compared to existing methods.
> - We have included **CIFAR-100 data in Tables 1, 2, and Figure 12**, added results for **networks with three hidden layers**, and provided PINN results for the **Diffusion and Poisson equations in Table 4**.
> - We conducted experiments in **Figure 6, 7,16 and Table 3** to evaluate whether the proposed method enables efficient learning with limited data. The results experimentally demonstrate that the proposed method is more data-efficient compared to other methods.
> - We conducted additional experiments, shown in **Figure 13 (a)**, to compare it with He initialization and Orthogonal initialization in ReLU neural networks.
> - Motivated by the improved performance of the proposed method in networks with significant variations in the number of nodes across layers, we further conducted experiments on **autoencoders, as presented in Figures 13 (b) and (c)**. These results demonstrate the applicability of the proposed method in such architectures.
>
> **Q3** *"Can the code used in the experiment can be provided to improve reproducibility?"*:
>
> A4: Thank you for this valuable suggestion. **Example code** implementing the proposed weight initialization method has been included in the Supplementary Material for reproducibility.
>
> We hope that these explanations address your concerns, but we'd be happy to answer any remaining questions about our method.
>
> ---
> [1] Karniadakis, George Em, et al. "Physics-informed machine learning." Nature Reviews Physics 3.6 (2021): 422-440.
>
> [2] Rathore, Pratik, et al. "Challenges in training PINNs: A loss landscape perspective." ICML 2024.
>
> [3] Gnanasambandam, Raghav, et al. "Self-scalable tanh (stan): Multi-scale solutions for physics-informed neural networks." TPAMI (20023).
>
> [4] Raissi, Maziar, Paris Perdikaris, and George E. Karniadakis. "Physics-informed neural networks: A deep learning framework for solving forward and inverse problems involving nonlinear partial differential equations."
> Journal of Computational physics 378 (2019): 686-707.

---

> > ### Comment · Reviewer_6yoV · 2024-11-27
> > **Thank you**
> >
> > I thank you for all your work in writing up the rebuttal and do apologize for the late response on my side. I am reading through the rebuttal and will provide some feedback soon.

---

> ### Author Response · Authors · 2024-11-25
> **Reminder for Reviewer 6yoV**
>
> Dear Reviewer 6yoV,
>
> We hope that our responses could adequately address your concerns. As the discussion phase deadline approaches, we warmly welcome further discussion regarding any additional concerns that you may have, and we sincerely hope you can reconsider the rating accordingly.
>
> Thank you for the time and appreciation that you have dedicated to our work.
>
> Best regards,
>
> Authors of submission 13420

---

> ### Comment · Reviewer_6yoV · 2024-11-27
>
> Although the proposed method has no significant improvement in classification tasks, the additional experiment in PINN shows consistent improvement across different settings when compared with a popular initialization method with Normalization. This improves the soundness of the paper, and I have increased the score accordingly.
>
> Quickly read through the references provided by author, some of them do not advocate for using tanh as an activation function, instead suggest for Swish. Some of them proposed a self-scalable version of it which may not suffer from the problem mentioned in the paper. A more detailed search reveals Swish, atan, tanh, elu are all popular choices of activation function and they are good at different problems. If my understanding is correct, the constraint imposed on the choosing activation function for PINN is only smoothness. This limits the contribution of this paper. Can the author comment on that?
>
> However, the reviewer believes the theoretical motivation for explaining an empirical observation and proposing a solution with theoretical backing is of interest to the community hence moving the score accordingly.
>
> The reviewer believes it is easier to criticize than to confirm excellence, the reviewer is not an expert in PINN so I also changed the score to reflect that.

---

> ### Author Response · Authors · 2024-11-27
>
> Thank you for your valuable feedback! We sincerely appreciate the time and effort you dedicated to reviewing our work.
>
> **W1** *"The proposed method has no significant improvement in classification tasks."*
>
> A1: The proposed initialization method is designed to effectively propagate input signals to deeper layers during the initial forward pass. Effective signal propagation ensures that different input signals remain distinct in deeper layers. While the performance difference compared to existing initialization methods may not be significant in classification tasks, the proposed method demonstrates **robustness to network size (Table 1, 2, and 4)** and **high data efficiency (Table 3, Figures 6, 7, and 16)**. We particularly highlight the data efficiency of our method as a notable advantage.
>
> **Q1** *Other activation functions.*
>
> A2: Thank you for your insightful comment! We acknowledge that PINNs use a variety of activation functions depending on the specific PDE problem, including:
> - PINNs with tanh activation function [1,2,3,4,5]
> - PINNs with swish activation function [6,7]
> - PINNs with locally adaptive activation functions [8]
> - PINNs with self-scalable tanh activation functions [9]
>
> Despite the diversity of activation functions, all the references above employ **Xavier initialization**. It is well-known that the effectiveness of an initialization method depends on the activation function [10, 11]. Xavier initialization was originally designed for networks using tanh activation function but is often applied to networks with other activation functions.  Proposed method was also designed for tanh networks, with the hypothesis that an initialization tailored for tanh would also perform well for other smooth activation functions. However, this remains a hypothesis.
> Unlike Xavier initialization, which is theoretically designed based on maintaining linearity at $x=0$ for activation functions, the proposed method is grounded in the fixed points of the tanh function, making it more dependent on the characteristics of tanh activation.
>
> In response to the reviewer's comment, we have revised the manuscript as follows:
> - We added experiments in **Figure 14 involving swish, elu and diffusion equation**.
> -  We added experimental results in **Figure 18**, comparing Xavier initialization and the proposed method for **PINNs with swish**.
>
> We hope that this response addresses your concerns, but we'd be happy to answer any remaining questions about our method. Thank you again for your thoughtful review.
>
> ***
> [1]Jin, Xiaowei, et al. "NSFnets (Navier-Stokes flow nets): Physics-informed neural networks for the incompressible Navier-Stokes equations."
> Journal of Computational Physics 426 (2021): 109951.
>
> [2] Rathore, Pratik, et al. "Challenges in training PINNs: A loss landscape perspective." arXiv preprint arXiv:2402.01868 (2024).
>
> [3] Son, Hwijae, Sung Woong Cho, and Hyung Ju Hwang. "Enhanced physics-informed neural networks with augmented Lagrangian relaxation
> method (AL-PINNs)." Neurocomputing 548 (2023): 126424.
>
> [4] Yao, Jiachen, et al. "Multiadam: Parameter-wise scale-invariant optimizer for multiscale training of physics-informed neural networks."
> International Conference on Machine Learning. PMLR, 2023.
>
> [5] Song, Yanjie, et al. "Loss-attentional physics-informed neural networks." Journal of Computational Physics 501 (2024): 112781.
>
> [6] Wang, Hongping, Yi Liu, and Shizhao Wang. "Dense velocity reconstruction from particle image
> velocimetry/particle tracking velocimetry using a physics-informed neural network." Physics of fluids 34.1 (2022).
>
> [7] Wang, Sifan, et al. "PirateNets: Physics-informed Deep Learning with Residual Adaptive Networks." arXiv preprint arXiv:2402.00326 (2024).
>
> [8] Cai, Shengze, et al. "Physics-informed neural networks for heat transfer problems." Journal of Heat Transfer 143.6 (2021): 060801.
>
> [9] Gnanasambandam, Raghav, et al. "Self-scalable tanh (stan): Multi-scale solutions for physics-informed neural networks."
> IEEE Transactions on Pattern Analysis and Machine Intelligence (2023).
>
>
> [10] He, Kaiming, et al. "Deep residual learning for image recognition." Proceedings of the IEEE conference on computer vision and pattern recognition. 2016.
>
> [11] Glorot, Xavier, and Yoshua Bengio. "Understanding the difficulty of training deep feedforward neural networks." Proceedings of the thirteenth international
> conference on artificial intelligence and statistics. JMLR Workshop and Conference Proceedings, 2010.

---

> > ### Comment · Reviewer_6yoV · 2024-11-28
> >
> > The reviewer acknowledges the data efficiency in PINNs adds to the soundness of the paper.

---

> > > ### Author Response · Authors · 2024-11-29
> > >
> > > We sincerely thank you for your thorough reviews and for your appreciation of our work.

---

### Official Review · Reviewer_79Su · 2024-10-29

**Soundness:** 3
**Presentation:** 3
**Contribution:** 3
**Rating:** 8
**Confidence:** 4

**Summary:**

The paper introduces a novel weight initialization technique specifically designed for neural networks using the tanh activation function. This technique is evaluated against the well-known Xavier initialization method using benchmark datasets. The experimental results demonstrate that the proposed initialization method enhances the convergence speed of Physics-Informed Neural Networks (PINNs) utilizing the tanh function, showing greater robustness to variations in network size. The findings indicate that the new initialization technique outperforms Xavier initialization in solving various problems related to Partial Differential Equations (PDEs).

**Strengths:**

The strength of this paper lies in the development of a novel weight initialization technique that facilitates faster convergence and enhances performance in physics-informed neural networks (PINNs) utilizing the tanh activation function.

**Weaknesses:**

1. The comparison of the proposed weight initialization technique solely with Xavier is insufficient; it should also be experimentally evaluated against other state-of-the-art weight initialization methods.
2. Using tanh activation function in entire neural network is not good practice that it has the drawback of the vanishing gradients for the very high and very low values of x.
3. The formulation is more complex than standard methods, which could complicate implementation as shown in Equation (1).
4. The optimal value of 𝛼 can be highly context-dependent, varying across different architectures, datasets, and tasks, which makes it less universally applicable. Additionally, the choice of 𝛼 can interact with other hyperparameters, such as learning rate and batch size, complicating the overall tuning process during backpropagation, as described in Equation (2).
5. In Section 4.1, the evaluation process utilizes three datasets—MNIST, FMNIST, and CIFAR-10—employing the tanh activation function in every layer. As shown in Table 2, as the number of hidden layers increases, loss gradually increases, which is indicative of overfitting. It would be more effective to use the proposed weight initialization in conjunction with state-of-the-art architectures for training deep neural networks.

**Questions:**

See above in weakness section.

---

> ### Author Response · Authors · 2024-11-23
>
> Thank you for your valuable comments! In the following section, we will address the weaknesses (W) and questions (Q) mentioned above. **The changes are marked in blue.**
>
> **W1** *"The comparison of the proposed weight initialization technique solely with Xavier is insufficient; it should also be experimentally evaluated against other state-of-the-art weight initialization methods."*:
>
> A1:  Thank you for this valuable suggestion. Recently, tanh neural networks have gained attention due to their use in PINNs [1]. Our goal is to propose a new initialization method for tanh neural networks that is robust to network size and improves data efficiency. Since tanh neural networks typically use Xavier initialization [1,2,3], we compared our method against it. Based on the reviewers' suggestions, we expanded our comparisons to include more methods.
> - We added **experiments using Xavier initialization with normalization methods** (Batch Normalization and Layer Normalization) in **Figure 4, Table 3, and Table 4**. The results demonstrate that the proposed method achieves faster convergence and improved robustness to network size compared to Xavier initialization with normalization.
> - To the best of our knowledge, Xavier initialization is the most commonly used and effective initialization for tanh neural networks. Therefore, we conducted **additional experiments, shown in Figure 13**, to compare it with He initialization and Orthogonal initialization in ReLU neural networks.
>
> **W2** *"Using tanh activation function in entire neural network is not good practice that it has the drawback of the vanishing gradients for the very high and very low values of x."*:
>
> A2: Thank you for your comment. Tanh activation is known to have higher computational complexity and gradient issues. However, it has been experimentally shown that tanh outperforms ReLU and other activation functions in Physics-Informed Neural Networks (PINNs) [1,4].
> - We **conducted additional experiments (Figure 13) comparing the performance of ReLU activation with He or Orthogonal initialization** against tanh neural networks with the proposed initialization. Despite the known issues with tanh activation, the results demonstrate that the proposed initialization enables tanh networks to achieve better performance.
> - We have added an **experiment in Figure 14 to evaluate the absolute error of PINNs based on different activation functions**. Based on the results of this experiment, we emphasize the importance of the tanh activation function in PINNs.
>
> **W3** *"The formulation is more complex than standard methods, which could complicate implementation as shown in Equation (1)."*:
>
> A3: Thank you for your comment. In Section 3.2, under Proposed Weight Initialization, the weight matrix is described as the sum of a matrix *D*, consisting of ones and zeros, and a noise matrix *Z*. While slightly more complex than Xavier initialization, it is expected to be straightforward to apply.
> -  We have enhanced **the clarity of the Proposed Weight Initialization in Section 3.2**.
> -  **Example code** implementing the proposed weight initialization method has been included in the Supplementary Material for reference.
>
> **W4** *"The optimal value of 𝛼 can be highly context-dependent, varying across different architectures, datasets, and tasks, which makes it less universally applicable. Additionally, the choice of 𝛼 can interact with other hyperparameters, such as learning rate and batch size, complicating the overall tuning process during backpropagation, as described in Equation (2)."*:
>
> A4: Thank you for your comment. Existing methods, such as Xavier initialization, He initialization, and Randomized Asymmetric Initialization [5], consider only the number of nodes in the hidden layers, without accounting for factors like datasets, learning rates, or batch sizes. In response to the reviewers’ comments, we have **added the following experiments**:
>
> - To investigate the dependency of 𝛼 on the dataset, we examined **Figures 3 and 8** and observed that datasets drawn from different distributions resulted in approximately normal distributions at specific layers.
> - To investigate the dependency of 𝛼 on the dataset, we conducted additional experiments with **CIFAR-100 in Tables 1 and 2**. We also extended **Table 4 to include experiments with the Diffusion and Poisson equations**.
> - To investigate the dependency of 𝛼 on architecture, we validated the method across **various network sizes and autoencoders, as shown in Table 2 and Figure 13**.
> - **Additional experiments were conducted in Table 3, Figure 6, Figure 7, and Figure 16** to investigate the dependency of
> 𝛼 on dataset size. These experiments further demonstrate the data efficiency of the proposed method compared to existing methods.

---

> ### Author Response · Authors · 2024-11-23
>
> **W5** *"In Section 4.1, the evaluation process utilizes three datasets—MNIST, FMNIST, and CIFAR-10—employing the tanh activation function in every layer. As shown in Table 2, as the number of hidden layers increases, loss gradually increases, which is indicative of overfitting. It would be more effective to use the proposed weight initialization in conjunction with state-of-the-art architectures for training deep neural networks."*:
>
> A5: Thank you for your thoughtful comments.
> - In **Table 2**, we conducted additional experiments on the CIFAR-100 dataset with a network containing three layers to better observe the trend of loss variation as the number of layers increases. The results show that the proposed method exhibits less overfitting in deep networks for feature-rich datasets such as CIFAR-10 and CIFAR-100.
> - We conducted **experiments on autoencoders with Batch Normalization and Dropout applied, as shown in Figure 13 (b) and (c)**. The proposed method demonstrated lower loss compared to other approaches.
>
> We hope that these explanations address your concerns, but we'd be happy to answer any remaining questions about our method.
> ***
> [1] Karniadakis, George Em, et al. "Physics-informed machine learning." Nature Reviews Physics 3.6 (2021): 422-440.
>
> [2] Rathore, Pratik, et al. "Challenges in training PINNs: A loss landscape perspective." ICML 2024.
>
> [3] Gnanasambandam, Raghav, et al. "Self-scalable tanh (stan): Multi-scale solutions for physics-informed neural networks." TPAMI (20023).
>
> [4] Raissi, Maziar, Paris Perdikaris, and George E. Karniadakis. "Physics-informed neural networks: A deep learning framework for solving forward and inverse problems involving nonlinear partial differential equations."
> Journal of Computational physics 378 (2019): 686-707.
>
> [5] Lu, Lu, et al. "Dying relu and initialization: Theory and numerical examples." arXiv preprint arXiv:1903.06733 (2019).

---

> ### Author Response · Authors · 2024-11-25
> **Reminder for Reviewer 79Su**
>
> Dear Reviewer 79su,
>
> We hope that our responses could adequately address your concerns.
> As the discussion phase deadline approaches, we warmly welcome further discussion regarding any additional concerns that you may have, and we sincerely hope you can reconsider the rating accordingly.
>
> Thank you for the time and appreciation that you have dedicated to our work.
>
> Best regards,
>
> Authors of submission 13420

---

> > ### Comment · Reviewer_79Su · 2024-11-26
> >
> > Thank you for submitting your revised paper. I appreciate the effort you've put into addressing the comments and improving the results. The updated work demonstrates significant progress, and I am impressed by the improvements. Congratulations on your efforts. Based on this revision, I am revising my rating to an 8. Keep up the great work, and I look forward to seeing how this research evolves.

---

> > > ### Author Response · Authors · 2024-11-26
> > >
> > > Dear Reviewer 79su,
> > >
> > > Thank you for your insightful and encouraging feedback. We sincerely appreciate the time and effort you dedicated to reviewing our work and for providing valuable suggestions that greatly contributed to improving the quality of our research. We are also deeply grateful for your recognition of our efforts and for revising your rating to an 8.
> > >
> > > Best regards,
> > >
> > > Authors of submission 13420

---

### Author Response · Authors · 2024-11-24
**Summary of Revisions**

Dear Reviewers and AC,

We would like to thank all the reviewers for taking the time to review our work and for providing valuable feedback.
We appreciate the recognition from reviewers of clear and good presentation **(Reviewers 79Su, 6yoV, kh6z, YCAx)**, improved performance across various tasks **(Reviewers 79Su, 6yoV, YCAx)**, the theoretical foundation of the proposed method **(Reviewers 6yoV, kh6z, YCAx)**, and the novelty of our method **(Reviewers 79Su, kh6z, YCAx)**.
***
The latest revision of our paper has been uploaded, addressing all comments and queries raised by the reviewers. Edits in the PDF have been highlighted in red. Below, we provide a summary of the changes made to our work.

# Writing
**Improvements** Thanks for the suggestions of Reviewers YCAx, and kh6z
- We have made **minor edits** for clarity and space efficiency.
- We have added **additional explanations in Section 3.1** for further clarification.
- We have added **inline headings** to each paragraph for improved organization.

# Experiments
**Normalization Methods** Thanks for the suggestions of Reviewer YCAx
- We have added experiments comparing **Xavier with normalization** and the proposed method on classification datasets in **Figure 4 and Table 3**.
- We have added an experiment comparing **Xavier with normalization** and the proposed method on PDEs in **Table 4**.

**Datasets Efficiency** Thanks for the suggestions of Reviewer 79Su.
- We have added experiments verifying **data efficiency** on classification datasets in **Table 3**.
- We have added experiments verifying **data efficiency** on PDEs in **Figure 6, 7 and 16**.

**Datasets and PDEs**  Thanks for the suggestions of Reviewer 6yoV, kh6z, and 79Su.
- We have added **CIFAR-100 data** to the existing experiments in **Tables 1 and 2**.
- We have added **the diffusion equation and Poisson equation** to the existing experiments in **Table 4**.
- We have added experiments on the **activation value distribution** in deeper layers in **Figures 3 and 8**.

**Supplementary Experiments**  Thanks for the suggestions of Reviewers 79Su, YCAx, 6yoV, and kh6z.
- We have added experiments on classification datasets between the proposed method for tanh and **He/orthogonal initialization for ReLU in Figure 13**.
- We have added an experiment on the **autoencoder in Figure 13 (b) and (c)**.
- We have added a performance comparison experiment of **PINNs based on various activation functions in Figure 10**.
- We have added an experiment on **PINNs with Swish** activation function in **Figure 18**.

We have included all experimental results in our revised paper.

Best regards,

Authors of submission 13420

---

### Comment · Area_Chair_W4E6 · 2024-11-25

Dear reviewers,

A reminder that **November, 26** is the last day to interact with the authors, before the private discussion with the area chairs. At the very least, please acknowledge having read the rebuttal (if present). If the rebuttal was satisfying, please improve your score accordingly. Finally, if you have concerns that might be solved in time, this is the last chance before moving on to the next phase.

Thanks,
The AC

---

### Meta-Review · Area_Chair_W4E6 · 2024-12-17

**Metareview:**

The paper proposes a novel initialization method for tanh neural networks, based on a fixed-point analysis of the layer.

The reviews are mixed, ranging from marginal rejection to strong acceptance. The reviewers had several technical questions (see below for a detail), but these were addressed in the rebuttal. They were also concerned about some missing baselines (e.g., other initialization techniques, layer normalization), which the authors added in the rebuttal. The only remaining concern is the limited scope of the paper, which focuses on a narrow type of neural network.

Overall, the paper is technically correct; the authors provided a very significant rebuttal which was appreciated by all reviewers who interacted during the rebuttal phase. In addition, they argue correctly that tanh models are still important in several sub-fields, such as PINNs. As a result, I lean towards acceptance.

**Additional Comments On Reviewer Discussion:**

- **Reviewer YCAx** had concerns on the mathematical analysis, and on the lack of experiments with layer normalization. These were addressed in the rebuttal.

- **Reviewer kh6z** had some questions on a few points (e.g., hyperparameters). However, the questions were rather vague, and the reviewer did not answer to my requests for clarification. They also ignored the author's rebuttal. As a result, I ignored the review in my final evaluation.

- **Reviewer 79Su** was originally negative, with concerns on the scope of the work (which is limited to tanh models), hyper-parameters, and the choice of baselines. However, the authors provided a significant rebuttal, and the reviewer now fully recommend acceptance. This was the most significant review for my final evaluation.

- **Reviewer 6yoV** had some minor concerns that were addressed in the rebuttal.

---

### Decision · Program_Chairs · 2025-01-22

Accept (Poster)